# Characteristics of aerosol vertical profiles in Tsukuba, Japan, and their impacts on the evolution of the atmospheric boundary layer

Rei Kudo[1], Toshinori Aoyagi[2], and Tomoaki Nishizawa[3]

[1]Meteorological Research Institute, Japan Meteorological Agency, Tsukuba, 305-0052, Japan
[2]Japan Meteorological Agency, Tokyo, 100-8122, Japan
[3]National Institute for Environmental Studies, Tsukuba, 305-0053, Japan

*Correspondence to*: Rei Kudo (reikudo@mri-jma.go.jp)

**Abstract.** Vertical profiles of the aerosol physical and optical properties, with a focus on seasonal means and on transport
events, were investigated in Tsukuba, Japan, by a synergistic remote sensing method that uses lidar and sky radiometer data.
The retrieved aerosol vertical profiles of the springtime mean and five transport events were input into our developed one-
dimensional atmospheric model, and the impacts of the aerosol vertical profiles on the evolution of the atmospheric
boundary layer (ABL) were studied by numerical sensitivity experiments. The characteristics of the aerosol vertical profiles
in Tsukuba are as follows: (1) The retrieval results in the spring showed that aerosol optical thickness at 532 nm in the free
atmosphere (FA) was 0.13, greater than 0.08 in the ABL owing to the frequent occurrence of transported aerosols in the FA.
In other seasons, optical thickness in the FA was almost the same as that in the ABL. (2) The aerosol optical and physical
properties in the ABL showed a dependency on the extinction coefficient. With an increase of the extinction coefficient from
0.00 to 0.24 km$^{-1}$, Ångström exponent increased from 0.0 to 2.0, the single-scattering albedo increased from 0.87 to 0.99,
and the asymmetry factor decreased from 0.75 to 0.50. (3) The large variability in the physical and optical properties of
aerosols in the FA were attributed to transport events, during which the transported aerosols consisted of varying amounts of
dust and smoke particles depending on where they originated (China, Mongolia, or Russia). The results of the numerical
sensitivity experiments using the aerosol vertical profiles of the springtime mean and five transport events in the FA are as
follows: (1) Numerical sensitivity experiments based on simulations conducted with and without aerosols showed that
aerosols caused the net downward radiation and the sensible and latent heat fluxes at the surface to decrease. The decease of
the temperature in the ABL (-0.2 to -0.6 K) and the direct heating of aerosols in the FA (0.0 to 0.4 K) strengthened the
capping inversion around the top of the ABL. Consequently, the ABL height was decreased by 133 to 208 m in simulations
with aerosols compared to simulations without aerosols. (2) We also conducted simulations in which all aerosols were
compressed into the ABL but in which the columnar properties were the same and compared with the simulation results for
uncompressed aerosol profiles. The results showed that the reductions in net downward radiation and in sensible and latent
heat fluxes were the same in both types of simulations. However, the capping inversion in the simulations with compression
was weakened owing to aerosol direct heating in the ABL and the lack of direct heating in the FA. This resulted in an
increase of the ABL height, compared with that in the simulations without compression. (3) The dependencies of the 2 m

temperature and ABL height on the optical thickness and Ångström exponent in the FA was investigated using the results of the numerical sensitivity tests. The 2 m temperature and ABL height was decrease with an increase of the optical thickness, and their decrease rates increase with a decrease of Ångström exponent because the optical thickness in the near infrared wavelength region was large when the Ångström exponent is small. However, there was a case that the Ångström exponent was large but the decrease of the ABL height was largest in all the simulation results. In this case, the strong capping inversion due to the large extinction coefficient around the top of the ABL was an import factor. These results suggest that the vertical profiles of the aerosol physical and optical properties, and the resulting direct heating has important effects on the ABL evolution.

# 1 Introduction

Solar radiation heats the Earth's surface, thereby causing thermal instability and evaporation. The thermal energy and water vapor are transported into the atmosphere through turbulent mixing in the atmospheric boundary layer (ABL). These processes in the ABL have important implications for global energy and water circulation. Aerosols have significant impacts on the radiation budget of the Earth because they scatter and absorb solar radiation (aerosol–radiation interaction) and modify cloud physical properties (aerosol–cloud interaction) (IPCC, 2013). In this study, we focus on aerosol–radiation interaction and did not consider clouds and precipitation processes. Direct scattering and absorption of solar radiation by aerosols decrease the amount of solar radiation that reaches the Earth's surface and sensible and latent heat fluxes, heat the atmosphere, and modify atmospheric stability. These effects have significant impacts on the evolution of the ABL, but the impacts differ depending on the aerosol optical properties (Yu et al., 2002; Pandithurai et al., 2008).

Yu et al. (2002) and Pandithurai, et al. (2008) investigated the influences of aerosol optical properties on the ABL structure by sensitivity experiments with a high resolution ABL model coupled with an accurate radiative transfer model. They showed that the light absorption characteristics of aerosols are important determinates of their impact on ABL evolution. However, these studies focused on aerosols only in the ABL. Tsunematsu et al. (2006) examined sounding data obtained by frequently launched sondes and showed that direct heating of transported dust in the free atmosphere (FA) strengthened the capping inversion at the top of the ABL. Therefore, it is also necessary to study the influences of aerosols in the FA on ABL evolution, especially because aerosols in the FA can be transported both regionally and globally (Uno et al., 2009).

Ground-based remote sensing has the advantage that it can be used for continuous monitoring of aerosol vertical profiles. We developed a synergistic method, SKYLIDAR, that retrieves vertical profiles of aerosol optical properties from lidar and sun/sky photometer data (Kudo et al., 2016). SKYLIDAR provides vertical profiles of the extinction coefficient, single-scattering albedo, and phase function, and with these products the solar heating rate can be evaluated (Kudo et al., 2016). Then, by inputting the retrieved aerosol optical properties into an ABL model, it is possible to investigate the influences of aerosols in the ABL and FA on the evolution of the ABL.

This study comprises two parts (Fig. 1). We first evaluated the vertical profiles of aerosol physical and optical properties in a two-year lidar and sun/sky photometer data set collected at Tsukuba, Japan, in a rural area located near the mega-city of Tokyo. The ABL height was determined from the lidar data to distinguish the locally emitted aerosols in the ABL and the transported aerosols in the FA, and the characteristics of the physical and optical properties in ABL and FA were investigated. The columnar properties of aerosols at Tsukuba have been investigated by many researchers (e.g., Nishizawa et al., 2004; Kudo et al., 2010a, 2010b, 2011), but these previous studies did not investigate their vertical profiles statistically. Second, we investigated the impact of aerosol vertical profiles on the evolution of the ABL by conducting numerical sensitivity experiments with our developed one-dimensional (1-D) atmospheric model which consists of the ABL and radiative transfer schemes. Details of the data set and methodologies are described in Sect. 2. The characteristics of the

aerosol vertical profiles and results of the sensitivity experiments conducted with the 1-D atmospheric model are presented in Sect. 3. Our findings are summarized in Sect. 4.

## 2 Data and methodology

### 2.1 Remote sensing of aerosol vertical profiles

#### 2.1.1 Data retrieval

The vertical profiles of aerosol optical and physical properties were estimated from sky radiometer and lidar data obtained by the SKYLIDAR remote sensing method (Kudo et al., 2016). The sky radiometer (Prede Co., Ltd, Tokyo, Japan), deployed in the SKYNET (Takamura and Nakajima, 2004), is a scanning photometer that measures direct solar radiation and the angular distribution of diffuse radiation. In this study, we used observation data at the Meteorological Research Institute (MRI) of

10 Japan Meteorological Agency (JMA) in Tsukuba (36.05°N, 140.12°E, about 25 m above sea level). Note that our data of the sky radiometer at Tsukuba is not transferred to the International SKYNET Data Center (http://www.skynet-isdc.org/index.php). The wavelengths of the sky radiometer data used in this study are 340, 380, 400, 500, 675, 870, and 1020 nm. We also used data from a two-wavelength Mie scattering lidar deployed by AD-Net (Sugimoto et al., 2015) at the National Institute for Environmental Studies (NIES) near MRI. The lidar data consisted of the attenuated backscatter

coefficients for particle and molecular scattering at 532 and 1064 nm, and the volume depolarization ratio including the contributions of particle and molecular depolarization at 532 nm. The sky radiometer and lidar observation data were collected during 2012 and 2013. As auxiliary data, we used vertical profiles of pressure and temperature from the U.S. National Centers for Environmental Prediction (NCEP) 6-hourly reanalysis data set (Kalnay et al., 1996), total ozone from observations made at the JMA Aerological Observatory (AO) near MRI, and surface albedo from the 5-year climatology of

the Filled Land Surface Albedo Product, which was generated from the official Terra/MODIS-derived Land Surface Albedo Product (Moody et al., 2005, 2007; Moody 2008). These auxiliary data were used for the calculation of Rayleigh scattering and gas absorption in the SKYLIDAR retrieval. The MRI, NIES and AO instruments are all located within a circle with a radius of 1 km.

SKYLIDAR estimates aerosol vertical profiles by the following two steps, based on a maximum a posteriori

scheme (Kudo et al., 2016). In the first step, the columnar values of the aerosol physical and optical properties (optical thickness, single-scattering albedo, etc.) are estimated by optimizing real and imaginary parts of the refractive index, volume size distribution, and volume ratio of non-spherical particles in the coarse mode to all of the sky radiometer data and the vertical mean of the depolarization ratio of the lidar data. Volume size distribution is assumed to follow a bi-modal lognormal distribution, and the volumes, mode radii, and widths of the fine and coarse modes are estimated. The optical

properties of non-spherical particles are calculated from a data table of randomly oriented spheroids (Dubovik et al., 2006). In the second step, the vertical profiles of the volume concentrations of fine and coarse modes, the volume ratio of non-

spherical particles in the coarse mode, and the real and imaginary parts of the refractive index are optimized to all of the lidar data and to the optical thickness and single-scattering albedo obtained in the first step. The final outputs are vertical profiles of the extinction coefficient, single-scattering albedo, phase function, the real and imaginary parts of the refractive index, the bi-modal volume size distribution, and the volume ratio of non-spherical particles in the coarse mode. The output wavelengths of the optical properties are 532 and 1064 nm. Note that the mode radii and width of the fine and coarse modes in the second step are fixed by the columnar values obtained in the first step. These outputs enable us to use the radiative transfer model to calculate the vertical profile of the solar heating rate (Kudo et al., 2016).

In the work of Kudo et al. 2016, we conducted the sensitivity tests of the SKYLIDAR using the simulated data of lidar and sky radiometer for the transported dust and pollution aerosol. The pollution aerosol was defined as small-sized and light-absorbing particles. The aerosol optical thickness at 500 nm in the simulation was from 0.05 to 1.2. The random errors were added to the simulated data. The errors were $\pm 2$ % for the direct solar radiation, $\pm 3$ % for the diffuse radiation, $\pm 10$ % for the attenuated backscatter coefficient, and $\pm 15$ % for the volume depolarization ratio. The SKYLIDAR successfully retrieved the columnar values (integrated values or vertical means) of the optical thickness, single-scattering albedo, asymmetry factor, real and imaginary parts of the refractive index, and volume size distribution in all the tests. However, the retrieval errors of the vertical profiles increased with a decrease of aerosol optical thickness. In the case that the aerosol optical thickness at 532 nm was 0.05, the retrieval errors were $\pm 0.003$ km$^{-1}$ for the extinction coefficient at 532 nm, and $\pm 0.05$ for the single-scattering albedo and asymmetry factor at 532 nm. The vertical profiles of the retrieved parameters other than the extinction coefficient had large oscillations due to the random errors of the lidar. In the sensitivity tests for the optical thickness more than 0.1, the vertical profiles of the size distribution, imaginary part of the refractive index, extinction coefficient, single-scattering albedo, and asymmetry factor of the transported dust were successfully retrieved, but that of the real part of the refractive index was not. In the transported pollution aerosol case, the vertical profiles of the size distribution, real part of the refractive index, extinction coefficient, and asymmetry factor were estimated well, but those of the imaginary parts of the refractive index and single-scattering albedo were not. When the SKYLIDAR failed to retrieve the vertical profiles of above mentioned parameters, the estimated vertical profiles were uniform, and the values were their vertical means.

## 2.1.2 Determination of ABL height

Locally emitted aerosols in the ABL and the transported aerosols in the FA can have different optical properties, which can be evaluated separately after the ABL height has been determined. We estimated the ABL height from the lidar data by the method of Baars et al. (2008), which is based on the wavelet covariance transform (WCT) with the Haar function. The WCT method is less affected by signal noise than the gradient and variance methods. The local maximum of the WCT vertical profile corresponds to the ABL height, and the local minimum corresponds to the base height of clouds or of transported aerosols. In this study, the ABL height was determined by the following procedure:

(1) The attenuated backscatter coefficients at 532 and 1064 nm were normalized by their maximum values below 1000 m, and the WCTs for data at 532 and 1064 nm were calculated.

(2) The local minima and maxima of the WCT vertical profiles at 532 and 1064 nm were detected.

(3) The base height of clouds (or transported aerosols) was searched by using a threshold of –0.1 for the local minimum of the WCT at 532 nm.

(4) If the base height was not detected in step (3), it was repeated using the WCT at 1064 nm. If the base height was still not detected, it was considered absent or unclear.

(5) The ABL height was searched by using a threshold of 0.05 for the local maximum of the WCT at 532 nm in the day-time, and of the WCT at 1064 nm at night. The top height of the search range was below the base height, if the base height of clouds or transported aerosols was detected in steps (3) or (4).

(6) If the ABL height was not detected, the threshold in step (5) was decreased by –0.01, and the search was repeated until the threshold reached to 0.01.

(7) If the ABL height was not detected in step (6), the search was repeated using the WCT at another wavelength.

(8) If the ABL height was not detected in step (7), the ABL height was considered undetermined.

(9) The time series of ABL height was smoothed by the running–mean with the time window of one hour.

Figure 2 shows the examples of the determined ABL height together with the extinction coefficient estimated by SKYLIDAR. The ABL height could be determined very well when transported aerosols were well above the ABL (Fig 2a). However, when transported aerosols become mixed with the aerosols in the ABL, the ABL height could not be detected (see from 0 to 7 UTC 2 April in Fig. 2b). In this case, we considered aerosols below base height to be in the ABL and those above to be in the FA. This assumption causes the uncertainties in evaluating the aerosol optical and physical properties in the FA and ABL. However, it is very difficult to evaluate the uncertainties because the mixing of the transported aerosols with those in the ABL makes the ABL height ambiguous, and the ABL height cannot be detected by the lidar data and our eyes. The successful retrievals of the ABL height by the above procedures from (1) to (9) was about 95 % of the 2,305 profiles under the clear sky conditions, and the base height was used as the ABL height in the remaining profiles. Therefore, the influences by our assumption would be small.

## 2.2 Model simulation

### 2.2.1 1-D atmospheric model

We developed a 1-D atmospheric model, consisting of ABL and radiative transfer (RT) schemes, and conducted sensitivity experiments to investigate the radiative impact of aerosols on the evolution of the ABL. The ABL scheme in the model is based on the ABL model used as the JMA operational mesoscale model for weather forecasting in Japan. The RT scheme is an RT model developed in our laboratory for the remote sensing of aerosols and clouds, and their impacts on the radiative balance in the solar and infrared wavelength regions (Asano and Shiobara, 1989; Nishizawa et al., 2004; Kudo et al., 2011).

The 1-D atmospheric model has a high resolution atmospheric vertical grid with 70 layers from the surface to 40 km. The thickness of the bottom layer is 5 m. Turbulent mixing is calculated by the Mellow-Yamada-Nakanishi-Niino Level 3 scheme (Nakanishi, 2001; Nakanishi and Niino, 2004, 2006), and calculations of surface fluxes are based on the Monin-Obukhov similarity using the universal function of Beljaars and Holtslag (1991). The vertical grid in the soil has 10 layers from the surface to 2 m depth, and the soil temperature is calculated by solving the diffusion equation. The water content in the soil layers was fixed in this study.

In the 1-D atmospheric model, vertical diffusion terms for turbulent mixing and vertical advection by a prescribed vertical motion field are considered for the vertical mixing of potential temperature, specific humidity and the horizontal component of wind. Neither cloud formation nor precipitation is included. In addition, vertical diffusion of aerosols is not considered in the model; aerosol vertical profiles are fixed by the initially given ones.

In the RT scheme, the solar spectrum from 300 nm to 3.0 μm and the infrared spectrum from 4. 0 to 50.0 μm are divided into 54 and 19 intervals, respectively. The downward and upward fluxes and the heating rate are calculated by the doubling and adding method (Lacis and Hansen, 1974). Gaseous absorption of water vapor, carbon dioxide, oxygen, and ozone are calculated by the correlated k-distribution method. Scattering at the ground surface is assumed to be Lambert reflection.

The aerosol parameters input to the RT scheme are the vertical profiles of the extinction coefficient, single-scattering albedo, and the phase function at wavelengths from 300 nm to 3.0 μm. However, the wavelengths of the SKYLIDAR retrievals are limited to 532 and 1064 nm. We determined the refractive index between 532 and 1064 nm by linear interpolation in a log-log plane and used the refractive index at 532 and 1064 nm for wavelengths of less than 532 nm and greater than 1064 nm, respectively (Kudo et al., 2016). The extinction coefficient, single-scattering albedo, and phase function from 300 nm to 3.0 μm were calculated from these refractive index, the volume size distribution, and the volume ratio of the non-spherical particles in the coarse mode. The influences of aerosols on the infrared wavelength region of more than 3.0 μm were ignored. The heating ratio estimated by this procedure was now validated now, but the surface solar radiation was compared with the measurements of the pyranometer. The difference was small, about 10 W m$^{-2}$ (Kudo et al. 2016).

### 2.2.2 Sensitivity experiments

We conducted three types of simulation experiments to investigate the impact of aerosols on the evolution of the ABL. The first type was simulations without aerosols (EXP0), the second was simulations using the observed aerosol vertical profile (EXP1), and the third was the same as the second one but with the entire aerosol vertical profile was compressed into the bottom 1 km (EXP2). Note that the columnar optical properties in EXP1 and EXP2 simulations were the same; only the vertical profile differed between them. Thus, the influences of aerosols can be evaluated from the difference between EXP0 and EXP1 simulations, and the influences of the aerosol vertical profile can be investigated by comparing the results of

EXP2 and EXP1 simulations. We conducted experiments using the springtime mean of the aerosol vertical profile and the aerosol vertical profiles observed in the spring during five aerosol transport events in the FA.

To set up the model parameter, we referred to the sensitivity experiments conducted by Yu et al. (2002) and Pandithurai et al. (2008). For our sensitivity experiments, we used the following specified parameters in the 1-D atmospheric model. The integration time of all simulations was 24 hours with a time step of 1 minute. The solar orientation was set to that on 5 April 2012 at 36.05°N. The surface albedo was set to the spring mean of the 5-year climatology of the Filled Land Surface Albedo Product (Moody et al., 2005, 2007; Moody 2008). The vertical motion was set to the spring mean of NCEP 6-hourly reanalysis data set. The initial vertical profiles of pressure, temperature, specific humidity, and horizontal wind were also set to the spring means of the NCEP 6-hourly reanalysis data set. The soil surface was assumed to be bare, and the heat capacity and thermal conductivity in the soil layers were set to $1.3 \times 10^6$ J m$^{-3}$ K$^{-1}$ and 0.3 W m$^{-1}$ K$^{-1}$, respectively, based on values for dry sandy clay (Kondo, 1994). The initial temperatures in the soil layers were based on the spring mean of the soil temperature observed at the weather observation field of Mito Meteorological Observatory (Ministry of Agriculture, Forestry, and Fisheries, and Japan Meteorological Agency, 1982), which is 60 km north of the MRI.

Because the sensible and latent heat fluxes at the surface depend on the water content of the soil, we performed sensitivity experiments for both dry and wet soils. The daily means of the sensible and latent heat fluxes in EXP0 for the dry soil case (volumetric water content 0.1) were 88 and 78 W m$^{-2}$, respectively. For the wet soil case (volumetric water content 0.2), the sensible heat flux was decreased by 22 W m$^{-2}$, and the latent heat flux was increased by 32 W m$^{-2}$. These differences affected the ABL structures (e.g., temperature and specific humidity) in the EXP1 and EXP2 experiments, but not the impacts of aerosols (i.e., the signs of differences, EXP1 – EXP0 or EXP2 – EXP0 were either both positive or both negative in the dry and wet soil cases, and their absolute values were not significantly different). Therefore, we do not show the results for the wet soil case in this paper. Thus, the volumetric water content in the soil layers was fixed at 0.1.

## 3 Results

### 3.1 Characteristics of aerosol vertical profiles

### 3.1.1 Seasonal characteristics

Frequency distributions of the extinction coefficient at 532 nm, based on daily means, were obtained for spring (43 analyzed days), summer (7 days), autumn (35 days), and winter (59 days) (Fig. 3). The small number in summer is due to the lack of completely clear-sky conditions. Summer in Japan is hot and humid, and the cumulous clouds develops almost every day. The SKYLIDAR can be applied to only the clear sky condition. In all the seasons, the extinction coefficient was large in the layer from the surface to 1.5 km altitude. This layer is the ABL, and the aerosols in this layer originate primarily from local emissions. In spring and winter, the two large peaks of the extinction coefficients were observed in the layers from 1.5 to 3.5 km and from 3.5 to 6 km altitude. These layers are in the FA, and most aerosols in these layers have been transported long

distances. Transported aerosols are frequently observed in the FA in spring, autumn, and winter, when low-pressure systems carrying aerosols emitted in the eastern region of the Eurasian continent frequently pass over Japan. In summer, Japan is dominated by a high-pressure system, so it receives fewer transported aerosols. In our data, the optical thickness in summer, autumn, and winter were almost the same in the ABL and FA, but in spring, optical thickness in the FA was 0.13, larger than

5 0.08 in the ABL (Table 1).

In general, the ABL height is high in summer and low in winter, but in our results, the ABL height was higher in winter and spring, and lower in summer and autumn (Fig. 3). The higher ABL heights in winter and spring can be attributed to the mixing of aerosols between the ABL and FA, which makes it difficult to determine the ABL height (see Sect. 2.1.2). The low ABL height in summer and autumn may be influenced by clouds, which form near the top of the convective mixed

layer. Under these circumstances, the ABL height cannot be determined from only lidar data.

The Ångström exponent is a parameter related to particle size: a smaller value indicates a larger particle size. We calculated the vertical profile of the Ångström exponent from the retrievals of the extinction coefficients at 532 and 1064 nm (Fig. 4). The Ångström exponent in the ABL was from 0.0 to 2.0 in all the seasons and increased as the extinction coefficient increased. This result suggests that large extinction coefficients were mainly due to small particles, such as sulfate, nitrate,

and organics. Conversely, background aerosols consist of large particles, such as locally emitted mineral dust, likely derived from the large areas of bare soil exposed by agriculture and urban development in Tsukuba. The Ångström exponent in the FA ranged from 0.0 to 2.5. This large variability can be attributed to differences in the composition of aerosols, in particular, the proportions of dust and smoke particles, during transport events. The characteristics of five transport events are described in Sect. 3.1.2.

Single-scattering albedo is an important parameter related to light absorption. In the FA, single-scattering albedo was around 0.95 with small variability (Fig. 5 and Table 1), but in the ABL, it was from 0.87 to 0.99 and decreased as the extinction coefficient decreased. In general, the single-scattering albedo of dust particles is small, whereas that of small particles, other than black carbon, is large (Hess et al. 1998, Aoki et al. 2005). The dependency of the single-scattering albedo on the extinction coefficient in the ABL is therefore consistent with the particle size result shown in Fig. 4.

The asymmetry factor is an indicator of how much solar energy reaches the surface: a large asymmetry factor value indicates strong forward scattering, which means that more solar energy reaches to surface. In addition, the value of the asymmetry factor is inversely proportional to that of the Ångström exponent. In our results, large variation of the asymmetry factor, from 0.4 to 0.8, was observed in the FA (Fig. 6). The asymmetry factor in the ABL was from 0.5 to 0.75, and a dependency of the asymmetry factor on the extinction coefficient was observed.

In Sect. 2.1.1, we described that the vertical profiles of the single-scattering albedo and asymmetry factor in the case of the small aerosol optical thickness, less than 0.1, contain large retrieval errors due to the signal noises of the lidar data. We should note that the single-scattering albedo and asymmetry factor where the extinction coefficient was less than 0.02 km$^{-1}$ in Figs. 5 and 6 might be contaminated with the retrieval errors.

The values of other important optical and physical parameters are shown in Table 1. These values are particularly useful for comparisons of aerosols in different areas. The values of the real and imaginary parts of the refractive index were from 1.41 to 1.45 and from 0.002 to 0.008, and they were similar values in both the ABL and FA and in all the seasons. When the SKYLIDAR fails to retrieve the vertical profiles of the real part of the refractive index in the transported dust case and the imaginary parts of the refractive index in the transported pollution case, the estimated vertical profiles are uniform and the values are their vertical means (Sect. 2.1). This may cause the similar values of the refractive index in the ABL and FA. The values of the mode radius were from 0.11 to 0.14 for the fine mode and from 2.83 to 5.89 for the coarse mode, respectively. They mostly did not differ among seasons, although the coarse mode radius was smaller in spring. The smaller coarse mode radius in spring reflects the relatively small coarse mode radius of the transported aerosols, which ranged from 1.93 to 3.61 μm (Table 2). In each season, the volume ratio of non-spherical particles in the coarse mode in the ABL was larger than that in the FA, owing to the presence of local dust in the ABL; the smallest value in the ABL was observed in summer, when the ground surface is generally covered with grasses and few dust particles are emitted from the surface. The lidar ratio (extinction to backscatter ratio) is an important parameter for estimating the extinction coefficient, particularly when only the lidar data are available for that purpose. We can calculate the lidar ratio from the single-scattering albedo and phase function in the SKYLIDAR retrievals. In our results, no clear seasonal difference was observed in the lidar ratio, and their values were around 60.

### 3.1.2 Aerosol transport events in the FA

The optical thickness in the FA was largest in spring among all seasons because of presence of transported aerosols. From our results obtained over two years, we selected for further examination five events, which occurred on 2 April 2012 and 16 April and 8, 9, and 14 May 2013, characterized by large optical thickness in the FA. The daily means of the optical and physical properties of transported aerosols in the FA on these five dates are summarized in Fig. 7 and Table 2. Large extinction coefficients were observed in the FA during these five transport events (Fig. 7a). In addition, we inferred that the aerosols during the events on 2 April 2012, 16 April and 14 May 2013 consisted primarily of transported dust, because on these dates the volume of coarse-mode particles was particularly large (Fig. 7b), Ångström exponent values were small, less than 1.0, and the volume ratio of non-spherical particles was large, from 0.84 to 0.99 (Table 2). The two-day backward trajectories for those events (Figs. 7c and d) suggest that the transported dust originated in desert areas of China and Mongolia. On 8 May 2013, the volume of fine-mode particles was very large (Fig. 7b), the Ångström exponent was also large, 1.82, and the volume ratio of non-spherical particles was smallest in all the cases (Table 2); these results indicate that the aerosols consisted dominantly of small and spherical particles. The backward trajectory (Figs. 7c and d) indicated that the source region was in Russia, to the southeast area of Lake Baikal, where a forest fire had been observed in early May 2013. Therefore, we interpreted this transported aerosol to consist of transported smoke particles were from that forest fire. The following day, 9 May 2013, the source had moved to northeastern China (Figs. 7c and d), and the volume of coarse-mode

particles was large (Fig. 7b); these results suggest that this aerosol may have consisted of transported smoke and dust particles.

The single-scattering albedo and asymmetry factor at 532 nm of the dust cases (2 April 2012, and 16 April and 14 May 2013), and the smoke and dust mixture case (9 May 2013) were from 0.95 to 0.98 and from 0.65 to 0.71, respectively (Table 2). Dubovik et al. (2002) summarized the AERONET retrievals in the world and showed the single-scattering albedo and asymmetry factor at visible wavelengths of 0.44 and 0.69 μm in the desert regions were from 0.92 to 0.98 and from 0.66 to 0.73, respectively. Moreover, the single-scattering albedo estimated from the sky radiometer for the Asian dust was from 0.91 to 0.97 (Uchiyama et al., 2005). These retrievals were the columnar values, but the cases that the coarse mode was dominant were selected. Our results were consistent with these results. The refractive index in this study was from 1.43 to 1.53 for the real part, and from 0.001 to 0.004 for the imaginary part, respectively. Aoki et al. (2005) summarised the refractive index of the dust from the reports of the various works, and showed that the real and imaginary parts at 500 nm are from 1.45 to 1.55, and from 0.0005 to 0.008, respectively. The mode radius for the coarse particles in this study was from 1.93 to 3.61 μm, and the AERONET retrievals in the desert regions were from 1.9 to 2.7 μm (Dubovik et al., 2002). The lidar ratio at 532 nm in this study was from 47 to 56, and the results of High Spectral Resolution Lidar or Raman lidar were from 20 to 70 (Burton et al., 2012; Groß et al., 2015). Consequently, our retrieved physical and optical properties of the transported dust were consistent with those reported in other studies.

For the transported smoke on 9 May 2013, the single-scattering albedo and asymmetry factor were 0.97 and 0.64, respectively. The SKYLIDAR fails to retrieve the vertical profile of the single-scattering albedo of the transported pollution aerosol (small-sized and light-absorbing particle). However, the estimated vertical profile is uniform, and the estimated value is the vertical mean (Sect. 2.1.1). Therefore, our estimated single-scattering albedo can be compared with that of the AERONET retrievals. Furthermore, since the extinction coefficient in the ABL was much smaller than that in the FA (Fig. 7a), the vertical mean of the single-scattering albedo would represent the transported smoke in the FA. The AERONET retrievals at visible wavelengths for the biomass-burning aerosols in Amazonian forest (Brazil), South American cerrado (Brazil), African savanna (Zambia), and Boreal forest (United Saes and Canada) were from 0.84 to 0.94 for the single-scattering albedo, and from 0.53 to 0.69 for the asymmetry factor (Dubovik et al., 2002). Our estimated asymmetry factor of 0.64 lied among these values, but the single scattering albedo of 0.97 was higher than the AERONET retrievals. In general, the smoke from the biomass-burning is composed of black carbon, organic carbon, and inorganic materials (Reid et al., 2005). The single-scattering albedo strongly depends on the fuel type and the burning conditions and ranges from 0.2 to 1.0 by depending on the ratio of black carbon (or elemental carbon) to organic carbon in the FLAME-4 experiment (Liu et al., 2013; Pokhrel et al., 2016). The AERONET retrievals for the boreal biomass-burning aerosols in the Alaska showed the single-scattering albedo in 2004 and 2005 was about 0.96 (Eck et al., 2009). They suggested a significant amount of smoldering combustion of woody fuels and peat/soil layers that would result in relatively low black carbon mass fractions for smoke particles. The black carbon fraction of the our analysed transported smoke also might be low. The refractive index of the smoke in this study was 1.42 for the real part and 0.003 for the imaginary part, respectively. These values were smaller

than those of the AERONET retrievals in above mentioned regions, from 1.47 to 1.52 for the real part and from 0.00093 to 0.021 for the imaginary part, respectively. Moreover, our results of the refractive index were similar to those of the water-soluble aerosols in the OPAC, which originates from gas to particle conversion and consists of various kind of sulfates, nitrates, and other, also organic, water-soluble substances (Hess et al., 1998). This supports the low black carbon fraction and large single-scattering albedo for the smoke in this study. It is possible that the black carbon fraction was decreased in the long-range transport from Russia to Japan by the increase of the water-soluble aerosols. The lidar ration of 61 at 532 nm in this study lied among the range of the observations by High Spectral Resolution Lidar or Raman lidar, from 50 to 100 (Burton et al. 2012; Groß et al. 2015).

## 3.2 Sensitivity experiment results

### 3.2.1 Impact of aerosols on the evolution of the ABL

Figure 8 and Table 3 show the results of EXP0 and EXP1 – EXP0. The net downward surface radiation in the solar and infrared wavelength regions, as well as the sensible and latent heat fluxes, were decreased in the EXP1 simulations (with aerosols) compared with EXP0 simulation (without aerosols) (Figs. 7a–c). The change in the daily mean ranged from -14 to -23 W m$^{-2}$ for the net downward radiation, from -7 to -11 W m$^{-2}$ for the sensible heat flux, and from -6 to -10 W m$^{-2}$ for the latent heat flux (Table 3). Absorption by the ground was also decreased; the change in the daily ranged from -1.3 to -2.2 Wm$^{-2}$ (Table 3). In general, the downward surface solar radiation becomes small when optical thickness is large, single-scattering albedo is small, and the asymmetry factor is small, (Kudo et al., 2010b). The single-scattering albedo and asymmetry factor were not very different between the springtime mean and the five transport events (Tables 1 and 2), so the reduction of the net downward radiation (Fig. 8a) mainly reflects the optical thickness of the column (Table 3), and the reductions of the sensible and latent heat fluxes were caused by the decrease of the net downward radiation. The potential temperature profile at noon local time decreased in the ABL owing to the decline in the sensible heat flux (Fig. 8d). Note that in the 1-D atmospheric model results, the latent heat flux could not warm the atmosphere in the ABL because condensation is not included in the model. The daily maximum 2 m temperature was decreased by 0.2 to 0.6 K (Table 3). In contrast, the potential temperature was increased by 0.0 to 0.4 K in the FA owing to the direct heating of transported aerosols (Fig. 8d). The vertical profiles of direct heating depended on the profiles of the extinction coefficient (Fig. 7a). The warming of the FA and the cooling of the ABL stabilized the atmosphere and strengthened the capping inversion around the top of the ABL. The strengthened capping inversion and the decline of the sensible heat flux caused the ABL height to decease by -133 to -208 m (Fig. 8f and Table 3).

The latent heat flux, that is, the water vapor flux, was apparently decreased by aerosols, but the change in the amount of surface evaporation was small, from -0.21 to -0.36 kg m$^{-2}$ day$^{-1}$ (Fig. 8c and Table 3). The change in the vertical profile of specific humidity was very small, but the specific humidity around the top of the ABL was decreased as a result of the decrease in the ABL height and the dry air in the FA (Fig. 8e).

**3.2.2 Impact of the aerosol vertical profile on the evolution of the ABL**

Figure 9 and Table 3 show the results of EXT0 and EXP2 – EXP0. Note that the entire aerosol vertical profile was compressed in the bottom 1 km in the EXP2 simulations, but the optical thickness of the column was the same as that in the EXP1 simulations. The influence of only the aerosol vertical profile can thus be investigated by comparing Figs. 8 and 9. The amounts of decrease in the net downward radiation and the sensible and latent heat fluxes in the EXP2 simulations were almost the same as those in the EXP1 simulations (Figs. 9a-c). However, the decrease in the potential temperature in the ABL was about -0.1 K at noon and was smaller in EXP2 than in EXP1 (Fig. 9d), because aerosol direct heating in the ABL was stronger in EXP2 than in EXP1. The changes in surface evaporation (Table 3) and specific humidity (Fig. 9e) in EXP2 were similar to those in EXP1. The aerosol direct heating in the ABL, together with the lack of direct heating in the FA, weakened the capping inversion around the top of the ABL. Therefore, the decrease in the ABL height was from -208 to - 133 m in EXP1 but those in EXP2 was from -90 to -24 m (Table 3). Thus, the evolution of the ABL was changed by the aerosol vertical profile, even though the columnar characteristics of the aerosol optical properties were the same. The impacts of aerosols on the ABL structure, that is, reductions of the temperature in the ABL and of the ABL height, were larger when aerosols were present in the FA.

**3.2.3 Relations between the aerosol physical and optical properties in the FA and the evolution of the ABL**

The relations between the aerosol physical and optical properties in the FA and the evolution of the ABL was investigated using the EXP1 results for the springtime mean and five transport events. Fig. 10 shows the dependencies of the decreases in the daily maximums of the 2 m temperature and ABL height on the optical thickness and Ångström exponent in the FA. We focused on the influences of Ångström exponent to the ABL evolution because the single-scattering albedo and asymmetry factor in the springtime mean and 5 events had similar values (Table 2). The solid lines in Fig. 10 are the simulation results for different optical thickness and Ångström exponent in the FA with the simplified aerosol vertical profile, where the vertical profiles of the aerosol physical and optical properties are uniform in the ABL (from the surface to 2 km altitude) and FA (from 2 to 6 km altitude), respectively. The physical and optical properties used in the simulations are summarized in Table 4 and are the means calculated from the results of the springtime mean and five transport events. The optical thickness and Ångström exponent in the FA were changed by using the different values of the total volume and volume ratio of the fine and coarse modes for the size distribution.

The 2 m temperature and the ABL height was decreased with an increase of the optical thickness (Fig. 10). This influence of aerosols was described in Sect. 3.2.1. We found that the decrease rates of the 2 m temperature and ABL height increased with a decrease of the Ångström exponent. The small value of Ångström exponent indicates the large optical thickness in the near infrared wavelengths region. Therefore, the large particles such as dust weakened the ABL evolution efficiently due to the influences for both the visible and near infrared wavelength regions. However, the plots for the springtime mean and five transport events in Fig. 10 were not completely consistent with the solid lines for the simulation

results because the aerosol vertical profiles used in the simplified simulations were different from those for the springtime mean and five transport events. Particularly, the decrease in the ABL height in 9 May 2013 (smoke and dust case) was larger than that in 2 April 2012 (dust case). This result is opposite to above mentioned influence of Ångström exponent. In both cases, the optical thickness in the FA was similar value, about 0.33. However, the geometric thickness of the aerosol layer in the FA was about 3 km, smaller 5 km in 9 May 2013 than in 2 April 2012, and the extinction coefficient from 1.0 to 3.5 km altitude was larger in 9 May 2013 than in 2 April 2012 (Fig. 7a). This resulted in the largest temperature increase in the FA and the strong capping inversion in 9 May 2013 (Fig. 8d). Consequently, the ABL height was low in 9 May 2013. The dependencies of the 2 m temperature and ABL height on the Ångström exponent were found in this study, but the most important factor was the vertical profile of the extinction coefficient, in particular the extinction coefficient around the top of the ABL.

## 4 Conclusion

We first investigated vertical profiles of aerosol physical and optical properties at Tsukuba, Japan, with focus on the seasonal means and on five aerosol transport events, by a synergistic remote sensing method (SKYLIDAR) using sky radiometer and lidar data in two years (2012 and 2013). Second, we investigated the impact of the aerosol vertical profile on the evolution of the ABL by conducting sensitivity experiments with our 1-D atmospheric model.

The vertical profiles of the seasonal mean extinction coefficients showed high loads of the locally emitted aerosols in the ABL, from the surface to 1.5 km altitude, and the transported aerosols in the FA, from 1.5 to 6 km altitude. In summer, autumn, and winter, the aerosol optical thickness was almost same in the ABL and FA. In spring, the optical thickness at 532 nm in the FA was 0.13 and was larger than 0.08 in the ABL.

The physical and optical properties of the aerosols in the ABL were dependent on the extinction coefficient: As the extinction coefficient increased from 0.02 to 0.24 $km^{-1}$, the Ångström exponent was increased from 0.0 to 2.0, the single-scattering albedo increased from 0.87 to 0.99, and the asymmetry factor decreased from 0.75 to 0.5. These characteristics suggests that the background aerosols consisted of the locally emitted dust particles, and the large extinction coefficient was attributed to an increase of the small and non-absorbing particles.

The optical and physical properties in the FA varied greatly owing to the presence of transported aerosols. We investigated the vertical profiles and backward trajectories of five transport events. In three events, the aerosol consisted of dust particles transported from desert regions of China and Mongolia. In one event, the aerosol consisted of small smoke particles transported from a forest fire in Russia. The aerosols of a fifth event consisted of both small and large particles which we interpreted as smoke and dust particles, respectively. The single-scattering albedo and asymmetry factor of the transported dust particles and mixture of dust and smoke particles were from 0.95 to 0.98 and from 0.65 to 0.71, respectively. These values were consistent with those reported in the other works for the Asian dust and the desert regions in the world. In the transported smoke case, the asymmetry factor was 0.64 and was consistent with the reports for the biomass-burning

aerosols in the world. However, the single-scattering albedo was 0.97 and was higher than the other reports. It is supposed that the black carbon fraction was low in the source of the smoke, or the black carbon fraction decreased in the long-range transport from Russia to Japan.

We conducted sensitivity experiments in which the aerosol vertical profiles of the springtime mean and the five transport events were input into our 1-D atmospheric model. The sensitivity experiments with (EXP1) and without aerosols (EXP0) showed that the aerosols decreased net downward surface radiation (-14 to -23 W m$^{-2}$) and sensible and latent heat fluxes (-7 to -11, and -6 to -10 W m$^{-2}$, respectively). These resulted in the decrease of the maximum 2 m temperature (-0.2 to -0.6 K). The decrease of the temperature in the ABL and the direct heating of the transported aerosols in the FA strengthened the capping inversion near the top of the ABL. Consequently, the ABL height was less developed in EXP1 simulations than in the EXP0 simulations, and the decrease of the ABL height due to aerosols was from -133 to -208 m.

To investigate the impact of only the aerosol vertical profile on the evolution of the ABL, we conducted simulations (EXP2) in which all aerosols were compressed into the ABL (0-1 km altitude), but in which the columnar optical thickness was the same as that in the EXP1 simulations. The net downward radiation and the sensible and latent heat fluxes were not changed, but the ABL height was increased, in EXP2 simulations compared with EXP1 simulations. This increase in the ABL height resulted from a weakened capping inversion caused by aerosol direct heating in the ABL and the lack of direct heating in the FA.

Using the results of EXP1 simulations for the springtime mean and five transport events, the dependencies of the decreases in the 2 m temperature and ABL height on the optical thickness and Ångström exponent in the FA were investigated. The 2 m temperature and ABL height were decreased with an increase of the optical thickness, and their decrease rates depended on Ångström exponent. The 2 m temperature and ABL height efficiently decreased in the case that the Ångström exponent was small because of the large optical thickness in the near infrared wavelength region. However, in the case of the smoke and dust mixture event, the Ångström exponent was large, but the decrease of the ABL height was largest in the springtime mean and five events. The extinction coefficient around the top of the ABL was largest in the five transport events, and the strong capping inversion resulted in the lowest ABL height.

These sensitivity experiment results suggest that the vertical profiles of the aerosol physical and optical properties, and resulting direct heating are essential factors in the evolution of the ABL. Moreover, it is particularly important to characterize aerosol optical properties in the FA because aerosols in the FA can be transported widely and therefore affect the ABL both regionally and globally.

Our 1-D atmospheric model did not consider cloud formation or precipitation, although both of these can be affected by aerosol-induced modification of atmospheric stability. In the future, we plan to develop a 1-D or 3-D model that includes these processes and investigate aerosol-cloud interactions by inputting the observed aerosol data into the models.

**5 Data availability**

The lidar data are available from the AD-Net (http://www-lidar.nies.go.jp). The sky radiometer data are available from the International SKYNET Data Center (http://www.skynet-isdc.org/index.php), but the sky radiometer data at Tsukuba, Japan are available on request by contacting the first author of the paper.

**Acknowledgements.**

This work was supported by the Japan Society for the Promotion of Science KAKENHI grant nos. 24510026, 15H01728, and 15H02808. NCEP reanalysis data were provided by the NOAA/OAR/ESRL PSD, Boulder, Colorado, USA, from its website at http://www.esrl.noaa.gov/psd/. The MODIS MCD12C1 product was retrieved from the online data pool, courtesy of the NASA EOSDIS Land Processes Distributed Active Archive Center (LP DAAC), USGS/Earth Resources Observation and Science (EROS) Center, Sioux Falls, South Dakota (https://lpdaac.usgs.gov/data_access/data_pool).

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

**Table 1.** Seasonal means and standard deviations of aerosol optical and physical properties.

| Physical and optical properties | | Spring | | Summer | | Autumn | | Winter | |
|---|---|---|---|---|---|---|---|---|---|
| | | ABL | FA | ABL | FA | ABL | FA | ABL | FA |
| Optical thickness[a] | | 0.08±0.03 | 0.13±0.08 | 0.07±0.02 | 0.07±0.05 | 0.05±0.02 | 0.05±0.02 | 0.05±0.02 | 0.06±0.04 |
| Ångström exponent | | 0.81±0.36 | 0.97±0.52 | 0.84±0.46 | 1.53±0.17 | 1.18±0.35 | 1.05±0.26 | 1.06±0.40 | 1.11±0.38 |
| Single scattering albedo[a] | | 0.93±0.03 | 0.96±0.01 | 0.92±0.05 | 0.92±0.06 | 0.96±0.03 | 0.95±0.03 | 0.96±0.02 | 0.96±0.03 |
| Asymmetry factor[a] | | 0.70±0.03 | 0.68±0.03 | 0.71±0.04 | 0.66±0.04 | 0.66±0.03 | 0.67±0.03 | 0.66±0.04 | 0.66±0.04 |
| Refractive index[a] | Real part | 1.44±0.05 | 1.46±0.04 | 1.41±0.03 | 1.41±0.02 | 1.42±0.04 | 1.41±0.03 | 1.42±0.05 | 1.42±0.03 |
| | Imaginary part | 0.006±0.004 | 0.003±0.002 | 0.006±0.004 | 0.008±0.006 | 0.003±0.002 | 0.003±0.002 | 0.002±0.002 | 0.002±0.002 |
| Mode radius (μm) | fine | 0.14±0.02 | | 0.14±0.04 | | 0.12±0.03 | | 0.11±0.02 | |
| | coarse | 2.83±1.45 | | 4.59±1.37 | | 4.70±2.00 | | 5.89±2.30 | |
| Mode width | fine | 0.46±0.13 | | 0.59±0.09 | | 0.53±0.13 | | 0.60±0.13 | |
| | coarse | 0.92±0.10 | | 0.98±0.02 | | 0.98±0.02 | | 0.97±0.06 | |
| Volume ratio of non-spherical particles in the coarse mode | | 0.96±0.06 | 0.85±0.22 | 0.79±0.20 | 0.68±0.21 | 0.95±0.08 | 0.91±0.09 | 0.97±0.07 | 0.86±0.14 |
| Lidar ratio[a] | | 69±10 | 58±7 | 68±23 | 65±13 | 57±10 | 63±10 | 56±9 | 56±10 |

[a]Wavelength is 532 nm.

| | | Spring | | Summer | | Autumn | | Winter | |
|---|---|---|---|---|---|---|---|---|---|
| | | ABL | FA | ABL | FA | ABL | FA | ABL | FA |

**Table 2.** Daily means of optical and physical properties of transported aerosols in the FA

| Physical and optical properties | | 2 April 2012 | 16 April 2013 | 8 May 2013 | 9 May 2013 | 14 May 2013 |
|---|---|---|---|---|---|---|
| Optical thickness at 532 nm | | 0.33 | 0.24 | 0.27 | 0.33 | 0.25 |
| Ångström exponent | | 0.49 | 0.47 | 1.82 | 1.28 | 0.78 |
| Single-scattering albedo at 532 nm | | 0.98 | 0.97 | 0.97 | 0.96 | 0.95 |
| Asymmetry factor at 532 nm | | 0.68 | 0.71 | 0.64 | 0.65 | 0.68 |
| Real part of the refractive index at 532 nm | | 1.53 | 1.43 | 1.42 | 1.53 | 1.48 |
| Imaginary part of the refractive index at 532 nm | | 0.001 | 0.001 | 0.003 | 0.004 | 0.004 |
| Mode radius (µm) | fine | 0.15 | 0.13 | 0.14 | 0.15 | 0.15 |
| | coarse | 2.43 | 2.28 | 2.15 | 3.61 | 1.93 |
| Mode width | fine | 0.31 | 0.46 | 0.43 | 0.44 | 0.48 |
| | coarse | 0.90 | 0.89 | 0.98 | 0.98 | 0.77 |
| Volume ratio of non-spherical particles in the coarse mode | | 0.99 | 0.97 | 0.34 | 0.96 | 0.84 |
| Lidar ratio at 532 nm | | 47 | 56 | 61 | 55 | 56 |

**Table 3:** Results of EXP0, EXP1, and EXP2 sensitivity experiments

| | Aerosol optical thickness in the column (532 nm) | Daily mean net downward radiation (W m$^{-2}$) | Daily mean sensible heat flux (W m$^{-2}$) | Daily mean latent heat flux (W m$^{-2}$) | Daily mean ground absorption (W m$^{-2}$) | Daily mean of 2 m temperature (K) | Daily maximum of 2 m temperature (K) | Daily integrated surface evaporation (kg m$^{-2}$ day$^{-1}$) | Daily maximum ABL height (m) |
|---|---|---|---|---|---|---|---|---|---|
| EXP0 | | | | | | | | | |
| Spring mean | 0.0 | 166 | 88 | 78 | -0.6 | 285 | 293 | 2.68 | 2352 |
| EXP1 – EXP0 | | | | | | | | | |
| Spring mean | 0.21 | -14 | -7 | -6 | -1.3 | -0.3 | -0.2 | -0.21 | -133 |
| 2 Apr 2012 | 0.38 | -22 | -10 | -10 | -2.2 | -0.5 | -0.6 | -0.34 | -186 |
| 16 Apr 2013 | 0.35 | -19 | -9 | -8 | -1.8 | -0.4 | -0.4 | -0.29 | -162 |
| 8 May 2013 | 0.32 | -18 | -8 | -8 | -1.7 | -0.4 | -0.4 | -0.27 | -150 |
| 9 May 2013 | 0.42 | -23 | -11 | -10 | -2.2 | -0.5 | -0.5 | -0.36 | -208 |
| 14 May 2013 | 0.37 | -22 | -11 | -10 | -2.1 | -0.5 | -0.5 | -0.33 | -163 |
| EXP2 – EXP0 | | | | | | | | | |
| Spring mean | 0.21 | -14 | -7 | -5 | -1.2 | -0.2 | -0.1 | -0.18 | -24 |
| 2 Apr 2012 | 0.38 | -22 | -11 | -9 | -1.9 | -0.4 | -0.2 | -0.30 | -72 |
| 16 Apr 2013 | 0.35 | -19 | -9 | -8 | -1.7 | -0.3 | -0.2 | -0.26 | -83 |
| 8 May 2013 | 0.32 | -18 | -9 | -7 | -1.6 | -0.3 | -0.2 | -0.25 | -77 |
| 9 May 2013 | 0.42 | -23 | -12 | -9 | -2.0 | -0.4 | -0.3 | -0.32 | -90 |
| 14 May 2013 | 0.37 | -22 | -11 | -9 | -1.9 | -0.3 | -0.2 | -0.30 | -70 |

**Table 4:** Physical and optical properties used in the simulations for the simplified aerosol vertical profiles in Fig. 10

| Physical and optical properties | | ABL (Surface to 2 km altitude) | FA (2 to 6 km altitude) |
|---|---|---|---|
| Optical thickness at 532 nm | | 0.083 | 0.1 to 0.4 |
| Ångström exponent | | 0.86 | 0.0 to 2.0 |
| Real part of the refractive index at all the wavelengths | | 1.45 | 1.48 |
| Imaginary part of the refractive index at all the wavelengths | | 0.005 | 0.003 |
| Mode radius (μm) | fine | 0.15 | 0.15 |
| | coarse | 2.5 | 2.5 |
| Mode width | fine | 0.40 | 0.40 |
| | coarse | 0.90 | 0.90 |
| Volume ratio of non-spherical particles in the coarse mode | | 0.97 | 0.82 |

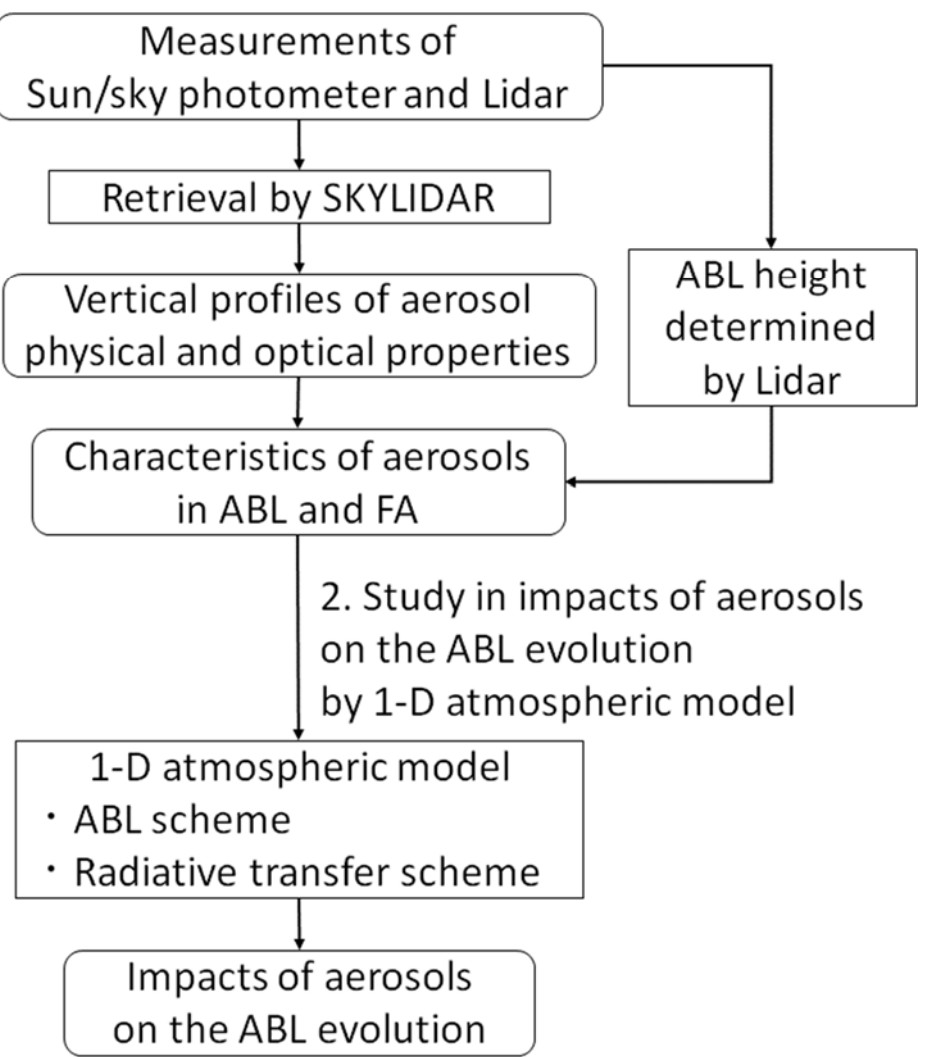

**Figure 1.** Flowchart diagram of this study.

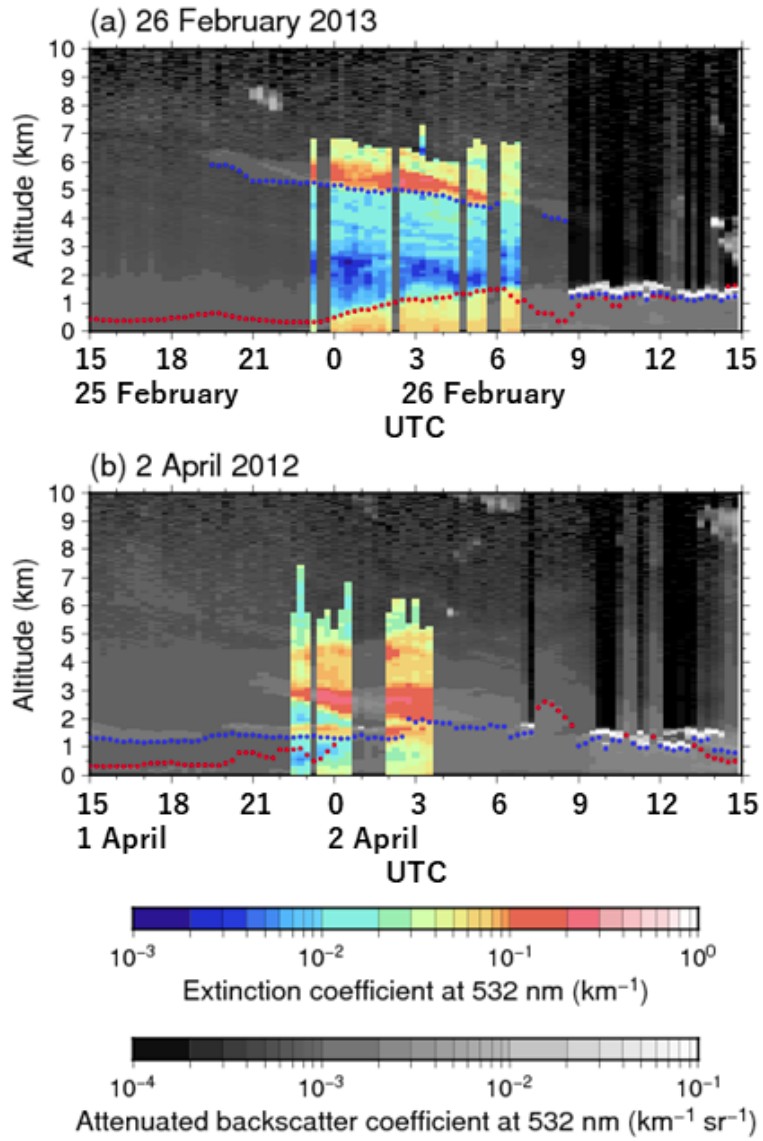

**Figure 2.** Two examples showing the determined ABL height (red dots) and the base height of clouds or transported aerosols (blue dots).

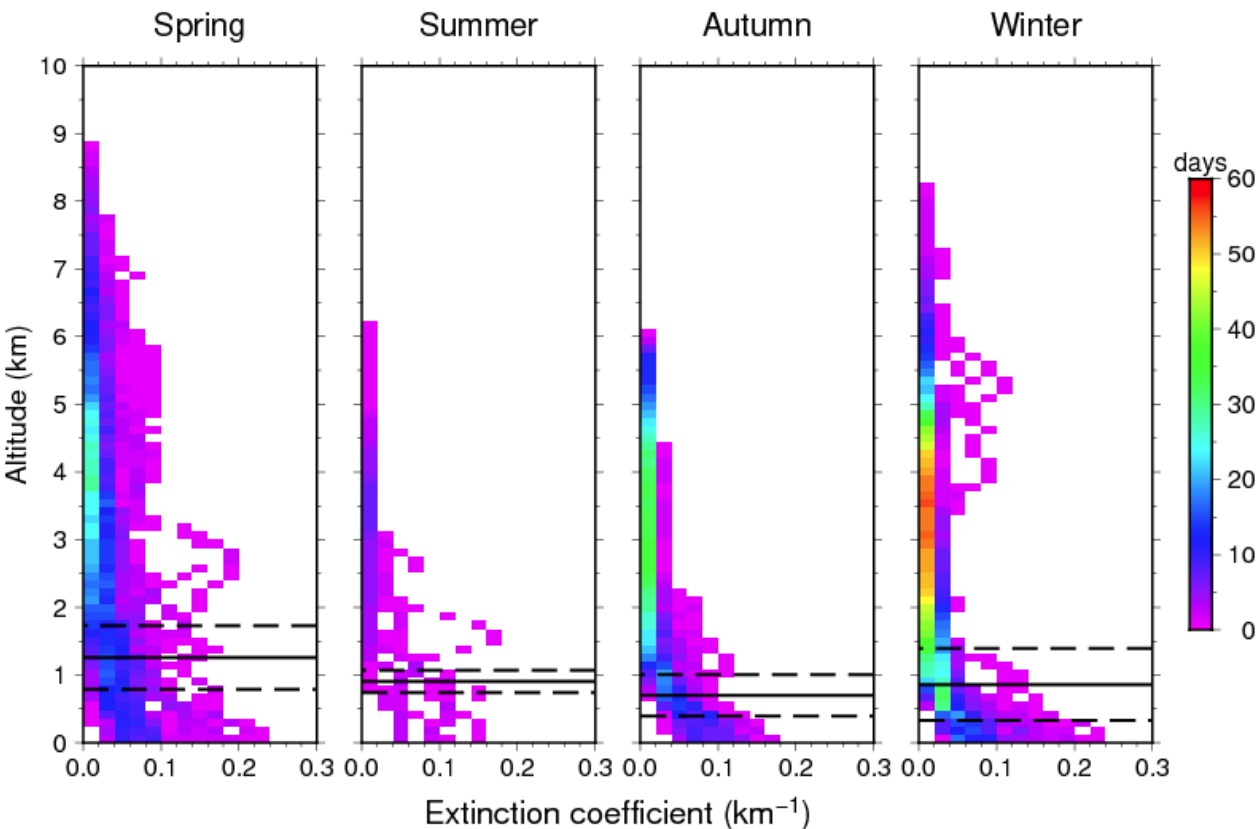

**Figure 3.** Frequency distributions of the extinction coefficient at 532 nm by seasons. The solid horizonal and dashed lines indicate the seasonal means and standard deviations of the ABL height.

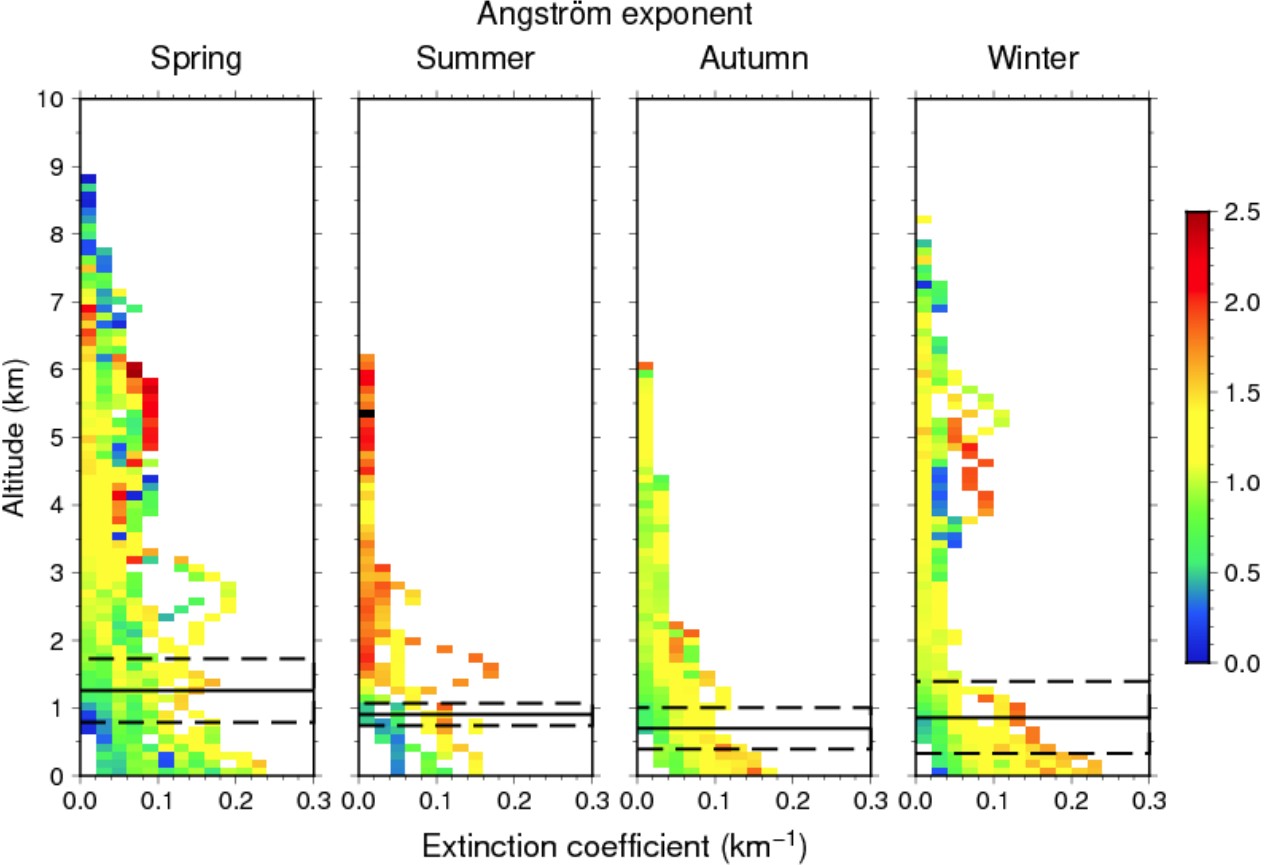

**Figure 4.** Dependencies of the Ångström exponent on the extinction coefficient at 532 nm and the altitude by seasons. The solid and dashed lines indicate the seasonal means and standard deviations of the ABL height.

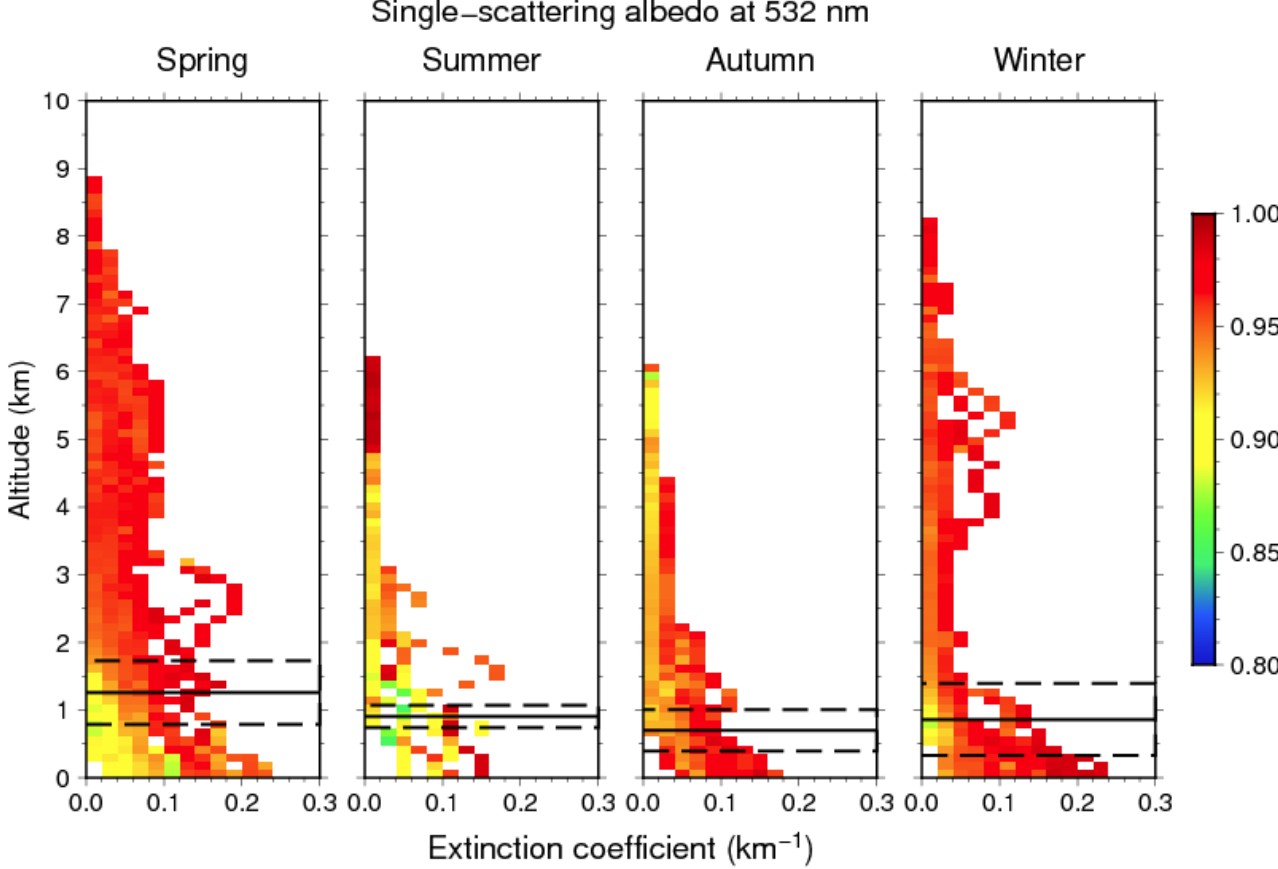

**Figure 5.** Dependencies of single-scattering albedo on the extinction coefficient at 532 nm and the altitude by seasons. The solid and dashed lines indicate the seasonal means and standard deviations of the ABL height.

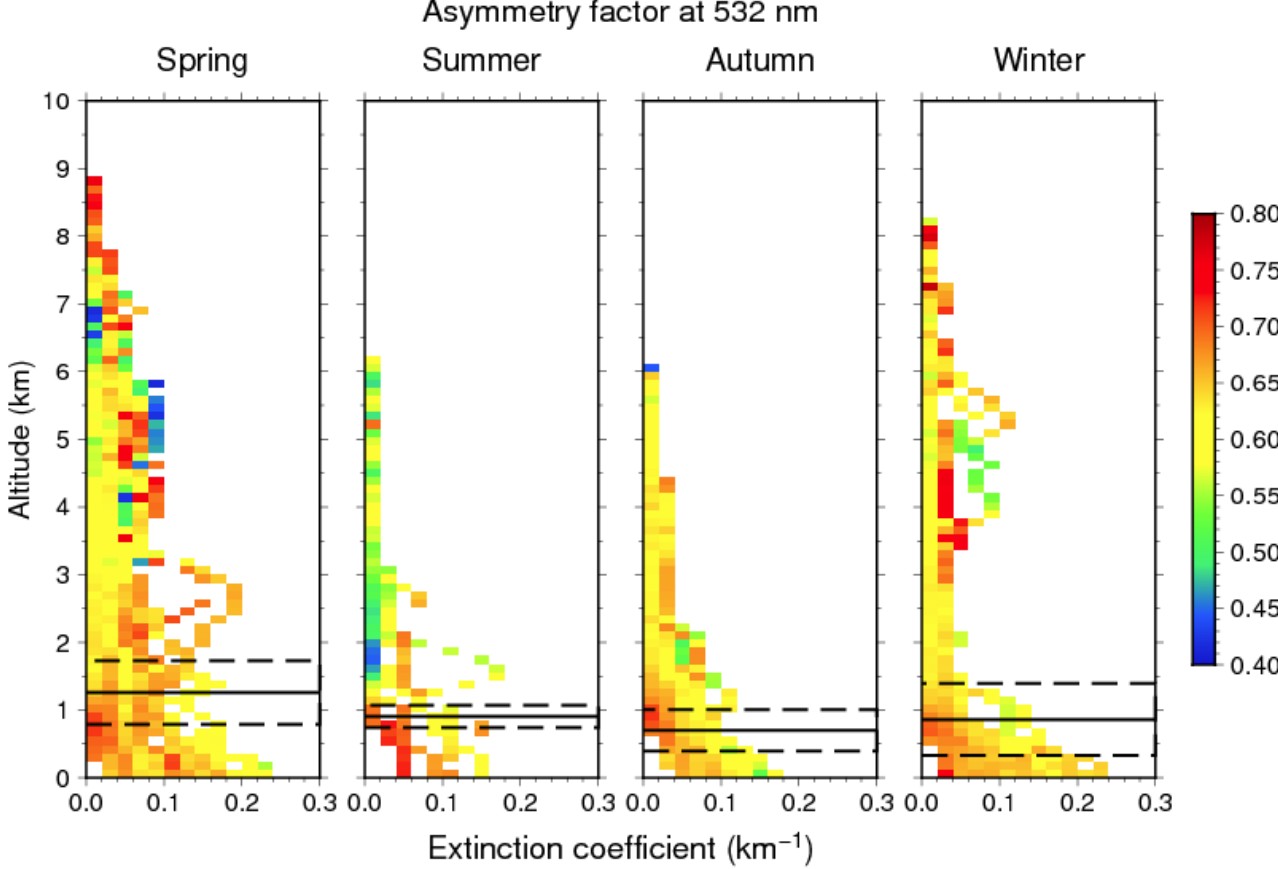

**Figure 6.** Dependencies of asymmetry factor on the extinction coefficient at 532 nm and the altitude by season. The solid and dashed lines indicate the seasonal means and standard deviations of the ABL height.

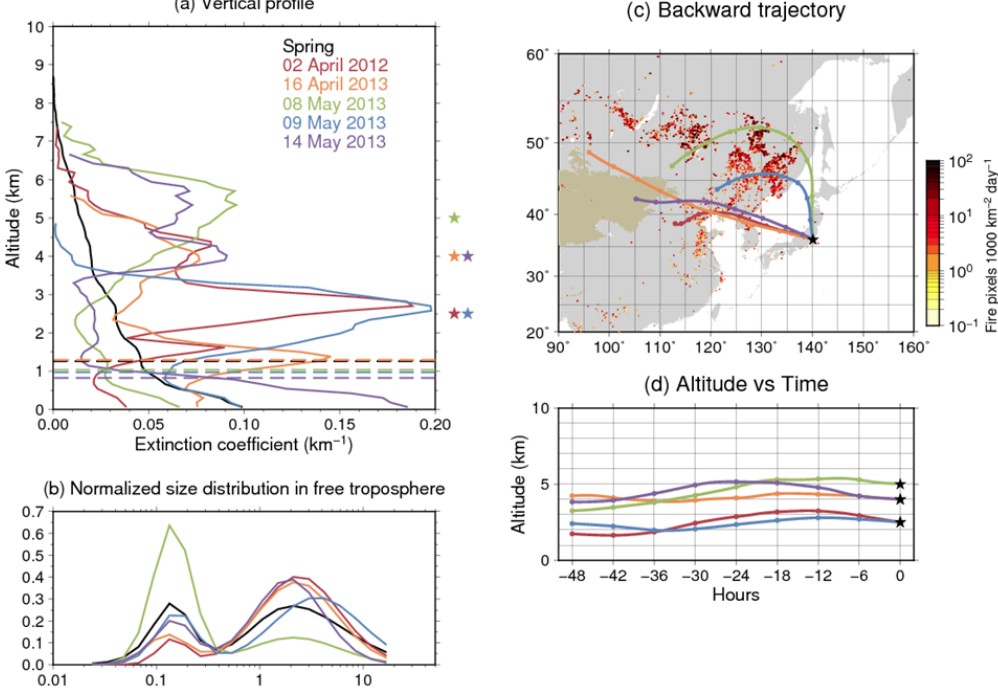

**Figure 7.** Optical and physical properties of transported aerosols in the FA: (a) vertical profile of the extinction coefficient at 532 nm with the ABL height (dashed lines), (b) normalized size distribution over the FA, (c) 2-day backward trajectory, and (d) altitude versus time cross section of the backward trajectory. The stars in (a) indicate the start altitude of the backward trajectories shown in (c). The color scale in (c) indicates fire activity from 1 to 9 May 2013, based on MODIS active-fire product data (NEO, 2016). The ochre color indicates desert regions, based on data of the Land Cover Type Climate Modeling Grid product (LP DAAC, 2013).

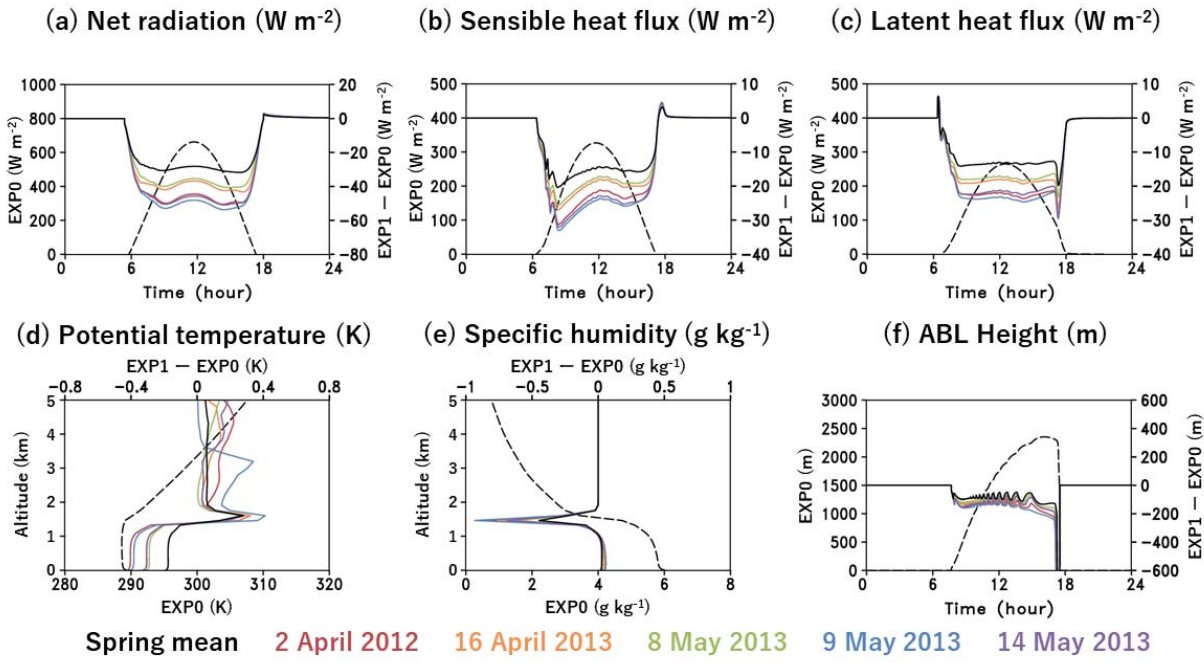

**Figure 8.** Results of EXP0 (dashed line) and the difference between EXP1 and EXP0 (solid lines): (a) net downward surface radiation, (b) sensible heat flux, (c) latent heat flux, (d) potential temperature at 12:00 LST, (e) specific humidity at 12:00 LST, and (f) ABL height.

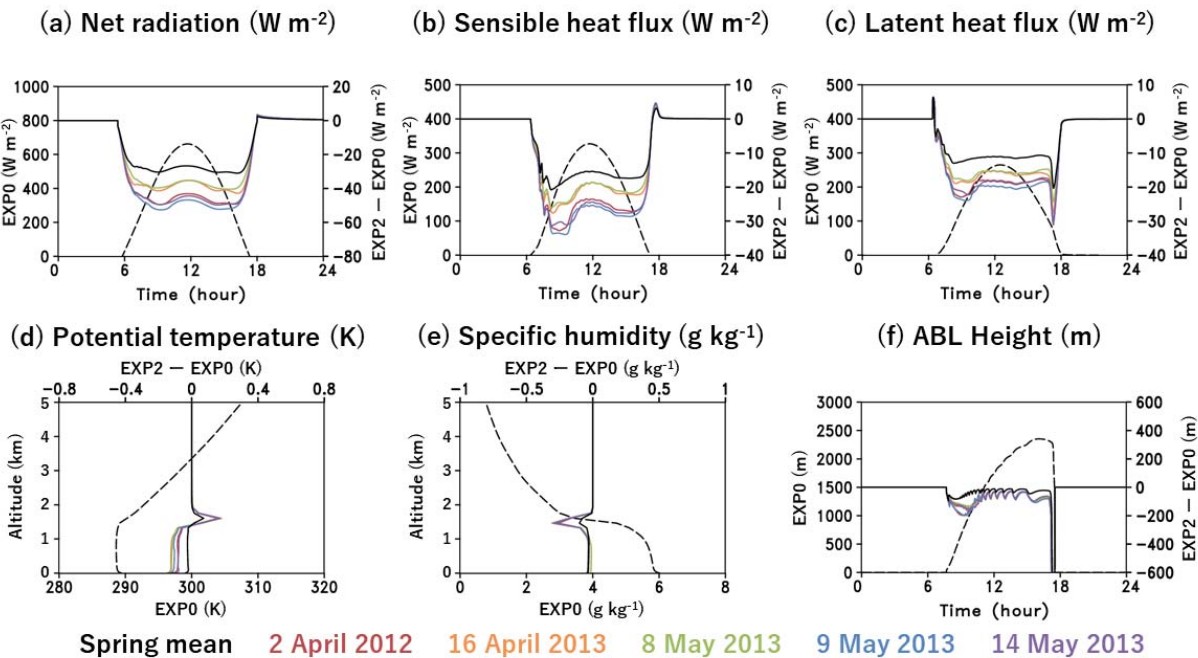

**Figure 9.** Results of EXP0 (dashed line) and the difference between EXP2 and EXP0 (solid lines): (a) net radiation at the surface, (b) sensible heat flux, (c) latent heat flux, (d) potential temperature at 12:00 LST, (e) specific humidity at 12:00 LST, and (f) ABL height.

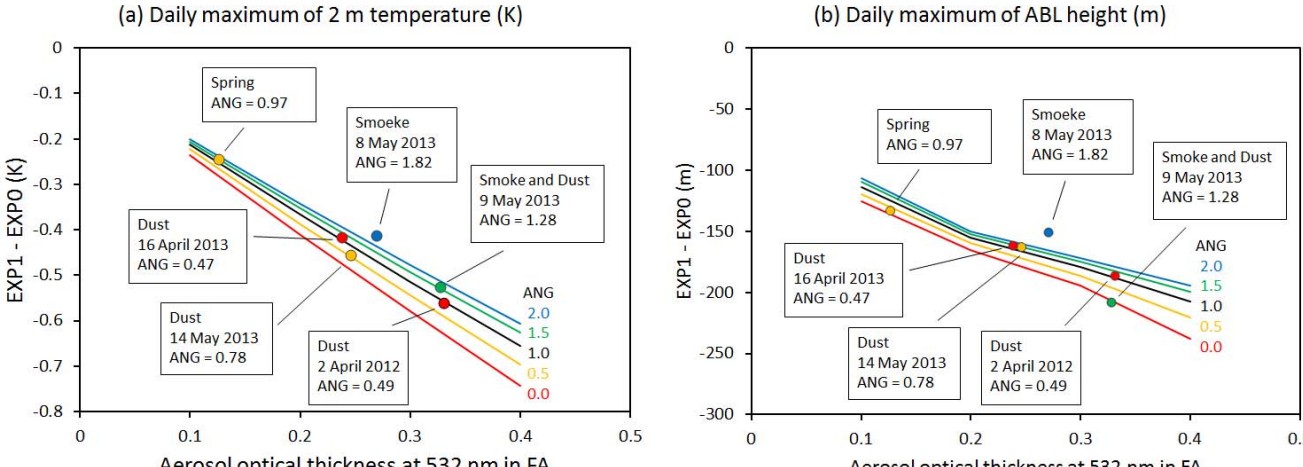

**Figure 10.** Dependencies of the daily maximum 2 m temperature (a) and the daily maximum ABL height (b) on the aerosol optical thickness and Ångström exponent in the FA. The "ANG" indicates Ångström exponent. The color of filled circle indicates the value of Ångström exponent, from 0.0 to 0.5 (red), from 0.5 to 1.0 (orange), from 1.0 to 1.5 (green), and from 1.5 to 2.0 (blue). The solid lines are the results of the model simulations for the simplified aerosol vertical profile described in the text.