# Peer review of "Characteristics of aerosol vertical profiles in Tsukuba, Japan, and their impacts on the evolution of the atmospheric boundary layer"

_Atmospheric Measurement Techniques, 2017_

## Referee Comment (RC1) · Anonymous Referee #1 · 7 Mar 2018

General Comments: This paper deals with rage resolved observations of aerosol properties, over Tsukuba, Japan, and a comprehensive study on how they affect the atmospheric boundary layer evolution. For the latter, the authors used the remote sensing observations as initial inputs in a 1-D atmospheric model. The paper rightly acknowledges previously related studies. The manuscript is well written, but in order to be improved, I would suggest to the authors to take into consideration the following comments. Minor Comments: 1. The Abstract section is well written. However, I would like to draw the attention to the authors to consider stating with numbers the main outcomes of their study. 2. Page 1, line 21: "compared to". 3. Page 1, lines 24-26: This sentence is too long. The authors are kindly requested to rephrase it,

and make their statement clearer. 4. Page 3, line 8: Apart from the coordinates, please provide also the elevation of Tsukuba station. It would be useful also for the reader, if you could provide the link of the used station which operates under SKYNET network. With a quick search I was not able to find this station here http://www.skynet-isdc.org/quicklooks.php. 5. Page 3, line 12: The authors are mentioning that among other aerosol properties the lidar data obtained by the AD-Net, contains also the depolarization ratio for particle and molecular scattering. Maybe the authors are referring to the physical quantity of volume depolarization ratio, which includes the contribution of molecular and particle depolarization. Please clarify this. 6. It is not so clear in the text, the contribution of the auxiliary data mentioned by the authors (page 3 lines 13-18). I suppose that this dataset was used in the radiative transfer module of the 1-D atmospheric model, but it would be nice if this is mentioned here. 7. Moreover, I would suggest to the authors to consider producing a flowchart diagram as the very first figure of their manuscript, in order to clearly demonstrate there the inputs and outputs of their approach/methodology. 8. Page 4 line 21: Define here the % of successful retrievals of ABL estimation from your dataset. 9. Page 4 line 23: What is the time window used for smoothing the time series of ABL height? 10. Page 7 line 4: Consider providing the appropriate reference for Ångström exponent namely: "Ångström, A., On the atmospheric transmission of Sun radiation and on dust in the air, Geogr. Ann., 11, 156– 166, 1929". Moreover, it is not clear, if this quantity refers to the entire atmospheric column, or is the range resolved backscatter or extinction related exponent calculated from the 532-1064 nm pair of wavelengths from the lidar measurements. Please try to clarify this in the manuscript. 11. Figure 3: Title "Ångström Exponent". Please go through the entire manuscript (tables, figures and text) to correct "Ångströme" to "Ångström". 12. Page 7 lines 14-15: The authors are kindly requested to provide some indicative studies related to the single scattering albedo of dust and black carbon. 13. Page 7 lines 29-31: The authors are commenting in lidar ratio values. But it is confusing since in 2.1.1 they mention that they operated a two wavelength Mie-scattering lidar from AD-Net. Since this system is not a Raman or

HSRL to obtain independently the lidar ratio, please comment from where these values came from. 14. Figure 6 (c): The authors are demonstrating two-days of air mass back-trajectories, in order to identify the aerosol source. Even though they provide some stars to indicate the starting altitude, the final height of the air mass arriving over the station as well as the height through the entire travel path is not shown. The authors are kindly requested to update this figure or at least mention these heights in the text. 15. I noticed that during the entire paragraph 3.1.2 the authors are describing qualitatively (e.g. "values were small; large; was large; was small" etc.) the aerosol optical and physical properties. They are kindly requested to provide also some numbers in the text. 16. Page 9 lines 5-6: Please specify also the warming range of the FA due to the direct aerosol heating. 17. Page 9 line 21: "...the decrease in the ABL height was smaller...". Please specify how much. 18. Figures 7 & 8: Please consider to make this graphs clearer. I think that it will help if: (1) units exist next to the parameters EXP0 and EXP1- EXP0 (2) provide altitude (y-axis in each graph) in km for being consistent with the previous figures. Finally I think that in the primary x-axis of (d) and (e) the x-title should be Potential Temperature (K) and Specific Humidity (g kg-1) respectively, and not EXP0. 19. Please update the link related to SKYNET, as provided in Page 11 line 3. Major Comments: 1. A general comment to the authors. Select some indicative numbers from the tables, representing the major findings of this study, and provide them into the manuscript (sections: Abstract, Results, Conclusions). 2. The authors apart from the mean statistical values are also using values observed from 5 case studies. The three of them where characterized as dust, one as smoke and one as mixture of dust and smoke. From their findings, the case studies differ only at the Ångström exponent and the aerosol load. I wonder if during their study, they can also draw any conclusion related to the influence of each individual aerosol type on the ABL evolution. 3. Page 4 lines 26-27: How much uncertainty this assumption may introduce? This maybe implies an overestimation of the aerosol load inside the ABL and an underestimation in the free troposphere. In any case the authors are kindly requested to comment on this. 4. Page 5 lines 21-22: Please provide an

estimation of the error that is introduced especially in the all wavelength net radiation that is discussed later, by ignoring the influence of aerosols on the infrared region ($\lambda$>1064nm), since dust particles are mostly in the coarse mode with diameters > 2 um. Moreover, the linear interpolation of the aerosol optical properties between 532 and 1064 nm can also influence the final retrievals. Please elaborate more on this. 5. Page 7 lines 2-3: The authors are giving an explanation of their findings regarding the seasonal variation of the mean ABL values, presented in Figure 2. However, I wonder if they compared these mean seasonal ABL values with corresponding measurements from radiosondes. In the same figure, it seems that there are a few cases that aerosol layers are detected up to 9 km height. Are these smoke particles advected from Russia? 6. Page 7 lines 23-25: Since particles located in FA and ABL may have totally different optical and chemical properties how comes and the refractive index is the same for the aforementioned atmospheric regions? Is this a bias coming from the retrieval method of SKYLIDAR? In any case the authors are kindly requested to comment more on this. 7. Page 8 line 15: Even though the aerosol type detected in 08 May2013, was characterized as smoke, no clear differences compared to the dust and dust smoke days can be found, regarding the values of single-scattering albedo and asymmetry factor (Table 2). Is this coming from the retrieval method of the SKYLIDAR?

Please also note the supplement to this comment:
https://www.atmos-meas-tech-discuss.net/amt-2017-462/amt-2017-462-RC1-supplement.pdf

———————————————

---

## Referee Comment (RC2) · Anonymous Referee #3 · 9 Mar 2018

General Comments

In this paper, the authors apply an inversion method that combines radiometric and lidar measurements and obtain vertical profiles of multiple aerosol optical and microphysical properties. The climatological behavior of these properties is then examined. Moreover, some of these products are used as input to an 1D atmospheric model in order to investigate the effect of the aerosol presence as well as their vertical distribution to the boundary layer height. In general the paper, is rather interesting and well organized. My main concern, however, is about the uncertainties of the inversion method applied, especially for cases where the aerosol load is small. Typically such inversion

methods require a certain level of confidence to the input parameters. This is usually achieved for the columnar properties when a sufficient amount of aerosol is present in the atmosphere. The lidar profiles, on the other hand, always include regions where the extinction coefficient is low, e.g. the last part of the profile. How accurate is the inversion for those cases? For example, are the retrievals for an extinction coefficient below 0.02 kmˆ-1 (Figures 2,3 and 4) trustworthy? This issues should be discussed in the manuscript. If the error of the retrieval has not been quantified, then the authors should consider applying thresholds to the AOD and extinction coefficients in order to exclude profiles, or parts of the profiles with potentially high uncertainties from their analysis.

Specific Comments

Section 2.1.2 (page 4, lines 11-12)

Why do you use the wavelet covariance transform (WCT) at 532nm instead of 1064nm for the cloud and aerosol base retrievals when both are available during the day? The aerosol layers tend to appear more clearly in the 1064nm channel.

Section 2.2.1 (page 5, lines 18-22)

It is mentioned here that "we determined the optical properties between 532 and 1064 nm by linear interpolation and used the optical properties at 532 and 1064 nm for wavelengths of less than 532 nm and greater than 1064 nm". Is this true for all the aforementioned optical properties? The aerosol extinction coefficient, in particular, usually has a strong wavelength dependence, even within the visible spectrum. Since the spectral range is much larger here, the authors should consider applying a correction, e.g. with the use of a constant angstrom exponent, for the extinction profiles in wavelengths below 532nm and above 1064nm.

Section 3.1.1 (page 6, lines 22-23)

The authors mention that "In all the seasons, the extinction coefficient was large in

three layers: from the surface to 1.5 km; from 1.5 to 3.5 km; and from 3.5 to 6 km altitude." The upper layer, however, only appears during winter and spring (Figure 2). Please rephrase. Since only the boundary layer is provided in the figures the discussion should be focused mainly on the differences between the free atmosphere and the boundary layer.

Section 3.1.1 (page 6, lines 31-32)

The number of daily mean profiles is considerably lower during summer (only 7 days - Figure 2). Is this related to cloudy or rainy conditions? The constant rainy conditions could explain the lower extinction values.

Section 3.1.2 (page 8)

Please provide specific values from the analysis (Table 2). A short comparison with estimates from other scientific studies would be interesting here as well.

Figure 4

The winter profiles exhibit high extinction and high SSA values while the summer profiles show low extinction and low SSA values. During spring both kinds of profile are observed. Do you observe any specific seasonal pattern within the spring months. For example, do the winter-like profiles tend occur in early spring while the summer-like profiles in late spring?

Technical Comments - Suggestions

Table 1

Consider defining abbreviations for the table properties in the caption and include the variability with +- in the same line with the mean value. Currently, each table property occupies 2 lines and sometimes this is difficult to follow.

Figure 6

A small height versus time plot per trajectory could be included along with the map. Regions where the trajectories pass closer to the ground are more probable to affect the layer aerosol content.

Section 3.1.1 (page 7, lines 14-15)

Add citation

Section 3.1.1 (page 7, lines 23-24)

typo - replace "almost same values" with "almost the same"

---

## Author Comment (AC1) · 27 Apr 2018

*General Comments: This paper deals with rage resolved observations of aerosol properties, over Tsukuba, Japan, and a comprehensive study on how they affect the atmospheric boundary layer evolution. For the latter, the authors used the remote sensing observations as initial inputs in a 1-D atmospheric model. The paper rightly acknowledges previously related studies. The manuscript is well written, but in order to be improved, I would suggest to the authors to take into consideration the following comments.*

Thank you very much for your time, evaluation and useful comments. We reflected your suggestions in the revised manuscript as well as the comments from the other reviewer, which improved the manuscript significantly. Point-by-point responses to your comments are written hereinafter.

Please note that we added two figures (Figs. 1 and 10), one table (Table 4) and Sect. 3.2.3 in the revised paper.

The changes to meet your comments are marked with blue in the revised manuscript.

*Minor Comments:*

*1. The Abstract section is well written. However, I would like to draw the attention to the authors to consider stating with numbers the main outcomes of their study.*

We revised the structure of the abstract.

*2. Page 1, line 21: "compared to".*

We corrected it.

Page 1, line 27 in the revised paper.

*3. Page 1, lines 24-26: This sentence is too long. The authors are kindly requested to rephrase it, and make their statement clearer.*

We revised it.

Page 1, lines 29-32 in the revised paper.

*4. Page 3, line 8: Apart from the coordinates, please provide also the elevation of Tsukuba station.*

*It would be useful also for the reader, if you could provide the link of the used station which operates under SKYNET network. With a quick search I was not able to find this station here http://www.skynetisdc.org/quicklooks.php.*

We added the elevation to the text.
Unfortunately, our data of the sky radiometer at Tsukuba is not transfer to the International SKYNET Data Center due to the policy of our institute. Therefore, the data at Tsukuba is not available online.
Page 4, lines 10-12 in the revised paper.

5. *Page 3, line 12: The authors are mentioning that among other aerosol properties the lidar data obtained by the AD-Net, contains also the depolarization ratio for particle and molecular scattering. Maybe the authors are referring to the physical quantity of volume depolarization ratio, which includes the contribution of molecular and particle depolarization. Please clarify this.*

"depolarization ratio for particle and molecular scattering" is "volume depolarization ratio" including the contribution of molecular and particle depolarization. We changed "depolarization ratio" to "volume depolarization ratio".
Page 4, lines 15-16 in the revised paper.

6. *It is not so clear in the text, the contribution of the auxiliary data mentioned by the authors (page 3 lines 13-18). I suppose that this dataset was used in the radiative transfer module of the 1-D atmospheric model, but it would be nice if this is mentioned here.*

These data were used for the calculation of Rayleigh scattering and gas absorption in the SKYLIDAR retrieval. These are necessary for the forward calculation of the sky radiometer and lidar data in the retrieval. We described this in the revised paper.
Page 4, lines 21-22 in the revised paper.

7. *Moreover, I would suggest to the authors to consider producing a flowchart diagram as the very first figure of their manuscript, in order to clearly demonstrate there the inputs and outputs of their approach/methodology.*

We added a flowchart (Fig. 1) to the revised manuscript.

Page 3, line 1, and page 24 in the revised paper.

8. *Page 4 line 21: Define here the % of successful retrievals of ABL estimation from your dataset.*

The successful retrievals of ABL estimation was about 95 % for 2,305 profiles under the clear sky conditions. In the 5 %, the base height of the transported aerosols was used as the ABL height. We added this to the manuscript.
Page 6, lines 23-24 in the revised paper.

9. *Page 4 line 23: What is the time window used for smoothing the time series of ABL height?*

One hour.
Page 6, line 15 in the revised paper.

9. *Page 7 line 4: Consider providing the appropriate reference for Ångström exponent namely: "Ångström, A., On the atmospheric transmission of Sun radiation and on dust in the air, Geogr. Ann., 11, 156– 166, 1929". Moreover, it is not clear, if this quantity refers to the entire atmospheric column, or is the range resolved backscatter or extinction related exponent calculated from the 532-1064 nm pair of wavelengths from the lidar measurements. Please try to clarify this in the manuscript.*

We are afraid that we were not able to get the paper of Ångström 1929 and cannot add the reference to the manuscript. We calculated the Ångström exponent from the retrieved extinction coefficients at 532 and 1064 nm. We added this explanation to the text.
Page 9, lines 11-13 in the revised paper.

10. *Figure 3: Title "Ångström Exponent". Please go through the entire manuscript (tables, figures and text) to correct "Ångströme" to "Ångström".*

We checked all the text and corrected.

11. *Page 7 lines 14-15: The authors are kindly requested to provide some indicative studies related to the single scattering albedo of dust and black carbon.*

We added the references to the revised paper.

Page 9, line 23.

12. *Page 7 lines 29-31: The authors are commenting in lidar ratio values. But it is confusing since in 2.1.1 they mention that they operated a two wavelength Mie-scattering lidar from AD-Net. Since this system is not a Raman or HSRL to obtain independently the lidar ratio, please comment from where these values came from.*

The lidar ratio was calculated from the retrieved single-scattering albedo and phase function. We added this explanation to the revised manuscript.

Page 10, lines 14-15 in the revised paper.

13. *Figure 6 (c): The authors are demonstrating two-days of air mass back-trajectories, in order to identify the aerosol source. Even though they provide some stars to indicate the starting altitude, the final height of the air mass arriving over the station as well as the height through the entire travel path is not shown. The authors are kindly requested to update this figure or at least mention these heights in the text.*

We added the altitude vs time plot as Fig. 7d to the revised manuscript.

Page 30.

14. *I noticed that during the entire paragraph 3.1.2 the authors are describing qualitatively (e.g. "values were small; large; was large; was small" etc.) the aerosol optical and physical properties. They are kindly requested to provide also some numbers in the text.*

We described our results with numbers and compared them with other studies in the revised manuscript.

Pages 10-12 in the revised paper.

15. *Page 9 lines 5-6: Please specify also the warming range of the FA due to the direct aerosol heating.*

The warming range is from 0.0 to 0.4 K at noon (Fig. 8d). We added this to the text.

Page 12, line 24 in the revised paper.

*16. Page 9 line 21: ". . .the decrease in the ABL height was smaller. . .". Please specify how much.*

We described the results with numbers in the revised manuscript.

Page 13, lines 9-11 in the revised paper.

*17. Figures 7 & 8: Please consider to make this graphs clearer. I think that it will help if: (1) units exist next to the parameters EXP0 and EXP1- EXP0 (2) provide altitude (y-axis in each graph) in km for being consistent with the previous figures. Finally I think that in the primary x-axis of (d) and (e) the x-title should be Potential Temperature (K) and Specific Humidity (g kg-1) respectively, and not EXP0.*

We revised the figures in accordance with your suggestions, (1) and (2). However, we were not able to delete "EXP0" from the x-title in the primary x-axis of (d) and (e) because we need to indicate that the primary x-axis is valid for only the result in EXP0.

Pages 31-32 in the revised paper.

*18. Please update the link related to SKYNET, as provided in Page 11 line 3.*

We updated the link.

Page 16, line 3.

*Major Comments:*

*1. A general comment to the authors. Select some indicative numbers from the tables, representing the major findings of this study, and provide them into the manuscript (sections: Abstract, Results, Conclusions).*

Thank you for your suggestion. We checked all the text and revised the descriptions.

*2. The authors apart from the mean statistical values are also using values observed from 5 case studies. The three of them where characterized as dust, one as smoke and one as mixture of dust and smoke. From their findings, the case studies differ only at the Ångström exponent and the aerosol load. I wonder if during their study, they can also draw any conclusion related to the influence of each individual aerosol type on the ABL evolution.*

We added a new section 3.2.3 and discussed the dependencies of the 2 m temperature and ABL height on the optical thickness and Ångström exponent. When the Ångström exponent was small (dust cases), the decrease rates of the 2 m temperature and ABL height were large (Fig. 10 in the revised paper). The dust particles have the small values of the Ångström exponent and the large optical thickness in the near infrared wavelength region. Therefore, the impacts of aerosols on the 2 m temperature and ABL height were large in the dust cases. However, the decrease of the ABL height in 9 May 2013 (smoke and dust case) was larger than that in 2 April 2012 (dust case). In both cases, the aerosol optical thickness in FA had similar values, about 0.33. The geometric thickness of the aerosol layer in the FA was smaller in 9 May 2013 than in 2 April 2012, and the extinction coefficient in the FA was larger in 9 May 2013 than in 2 April 2012 (Fig. 7a in the revised paper). Therefore, the temperature increase in the FA was large in 9 May 2013 (Fig. 8d in the revised paper), the capping inversion became strong, and the ABL height was less developed. We found the dependencies of the 2 m temperature and ABL height on the Ångström exponent, but the most important factor was the vertical profile of the extinction coefficient above the ABL.
Pages 13-14 in the revised paper.

*3. Page 4 lines 26-27: How much uncertainty this assumption may introduce? This maybe implies an overestimation of the aerosol load inside the ABL and an underestimation in the free troposphere. In any case the authors are kindly requested to comment on this.*

As you mentioned, our method may overestimate the aerosol load inside the ABL around 3 UTC in Fig. 2b in the revised paper. Our assumption can affect the results of aerosol optical properties in ABL and FA. However, it is very difficult to determine the ABL height of the case that the aerosols in the ABL and FA are mixed. Since we do not know the true ABL height, we cannot evaluate the uncertainties of our method. However, the number of the case, in which the base height was used as the ABL height, was 5 %. We believe the influence of our assumption would be small. I added this explanation in the revised paper.
Page 6, lines 20-25 in the revised paper.

*4. Page 5 lines 21-22: Please provide an estimation of the error that is introduced especially in the all wavelength net radiation that is discussed later, by ignoring the influence of aerosols on the infrared region (λ>1064nm), since dust particles are mostly in the coarse mode with diameters > 2 um. Moreover, the linear interpolation of the aerosol optical properties between 532 and 1064 nm can also influence the final retrievals. Please elaborate more on this.*

We consider the influences of aerosols in the wavelength region from 300 nm to 3.0 um. We calculated the real and imaginary parts of the refractive index between 532 and 1064 nm by the linear interpolation in a log-log plane. For wavelengths less than 532 nm and greater than 1064nm, the refractive index at 532 and 1064 nm were used. The extinction coefficient, single-scattering albedo, and phase function from 300 nm to 3.0 um were calculated from these refractive indices, the volume size distributions, and the volume ratio of the non-spherical particles in the coarse mode. Then, the radiative fluxes and heating rates were calculated by the radiative transfer model. We revised the text. Page 7, lines 16-25 in the revised paper.

*5. Page 7 lines 2-3: The authors are giving an explanation of their findings regarding the seasonal variation of the mean ABL values, presented in Figure 2. However, I wonder if they compared these mean seasonal ABL values with corresponding measurements from radiosondes. In the same figure, it seems that there are a few cases that aerosol layers are detected up to 9 km height. Are these smoke particles advected from Russia?*

We determined the ABL height from the lidar data in order to distinguish the locally emitted aerosols and the transported aerosols. For this purpose, we should evaluate the ABL height from the lidar data, which directly relates to the aerosol vertical profile. The ABL height estimated from the radiosonde may not be consistent with the aerosol vertical distribution. Therefore, we did not compare the ABL height with that by radiosonde. Moreover, the radiosonde launched only twice in a day (09 and 21 JST. JST is Japan Standard Time.) in Japan. The ABL height around the noon cannot be obtained from the radiosonde.

The smoke from Russia in 8 May 2013 was not detected near 9 km height (Fig. 7a in the revised paper). The aerosols around 9 km is sometimes observed in Tsukuba, Japan. However, we do not know the source of aerosols at 9 km altitude because our backward trajectory analysis was conducted for only the five cases of high aerosol load in the FA. In the future study, we will calculate a lot of backward trajectories for investigating the statistical relations between the aerosol optical properties and their

sources.

6. *Page 7 lines 23-25: Since particles located in FA and ABL may have totally different optical and chemical properties how comes and the refractive index is the same for the aforementioned atmospheric regions? Is this a bias coming from the retrieval method of SKYLIDAR? In any case the authors are kindly requested to comment more on this.*

In the work of Kudo et al. 2016, we conducted the sensitivity tests using the simulated data of lidar and sky radiometer for the transported dust and pollution cases to investigate the capability of the SKYLIDAR. The pollution aerosol was defined as small-sized and light-absorbing particle. The columnar values of the refractive index were estimated well in both cases. In the transported dust case, the vertical profile of the imaginary part of the refractive index was estimated successfully, but the that of the real part was not. In the transported pollution case, the vertical profile of the real part was estimated, but that of the imaginary part was not. In the failed cases, the vertical profile of the refractive index was constant with their vertical means. Therefore, the similar values of the refractive index in the ABL and FA may be caused by the errors of the SKYLIDAR. I added these explanation to the text. Page 5, lines 8-25, and page 10, lines 2-8 in the revised paper.

7. *Page 8 line 15: Even though the aerosol type detected in 08 May2013, was characterized as smoke, no clear differences compared to the dust and dust smoke days can be found, regarding the values of single-scattering albedo and asymmetry factor (Table 2). Is this coming from the retrieval method of the SKYLIDAR?*

In the numerical sensitivity tests of the SKYLIDAR (Kudo et al. 2016), the vertical profiles of single-scattering albedo and asymmetry factor for the transported dust were estimated well. Our retrievals in the transported dust cases were from 0.95 to 0.98 for the single-scattering albedo and from 0.65 to 0.71 for the asymmetry factor. These were consistent with the reports of the AERONET retrievals in the desert regions in the world, and the sky radiometer retrievals for the Asian dust.

The SKYLIDAR fails to retrieve the vertical profile of the single-scattering albedo of the transported smoke, but the estimated vertical profile is constant with the vertical mean (Kudo et al. 2016). Therefore, our estimated single-scattering albedo can be compared with that of the AERONET retrievals. The comparisons with other studies showed that the our estimated single-scattering albedo and asymmetry factor lied among those reported in the other studies. We added the explanation of the retrieval errors of the SKYLIDAR and the comparisons of our retrievals with other studies to the

revised paper.

Pages 10-12 in the revised paper.

[revised manuscript text omitted]

---

## Author Comment (AC2) · 27 Apr 2018

Thank you very much for your time, evaluation and useful comments. We reflected your suggestions in the revised manuscript as well as the comments from the other reviewer, which improved the manuscript significantly. Point-by-point responses to your comments are written hereinafter.

Please note that we added two figures (Figs. 1 and 10), one table (Table 4) and Sect. 3.2.3 in the revised paper.

The changes to meet your comments are marked with blue in the revised manuscript.

*General Comments*

*In this paper, the authors apply an inversion method that combines radiometric and lidar measurements and obtain vertical profiles of multiple aerosol optical and microphysical properties. The climatological behavior of these properties is then examined. Moreover, some of these products are used as input to an 1D atmospheric model in order to investigate the effect of the aerosol presence as well as their vertical distribution to the boundary layer height. In general the paper, is rather interesting and well organized. My main concern, however, is about the uncertainties of the inversion method applied, especially for cases where the aerosol load is small. Typically such inversion methods require a certain level of confidence to the input parameters. This is usually achieved for the columnar properties when a sufficient amount of aerosol is present in the atmosphere. The lidar profiles, on the other hand, always include regions where the extinction coefficient is low, e.g. the last part of the profile. How accurate is the inversion for those cases? For example, are the retrievals for an extinction coefficient below 0.02 km^-1 (Figures 2,3 and 4) trustworthy? This issues should be discussed in the manuscript. If the error of the retrieval has not been quantified, then the authors should consider applying thresholds to the AOD and extinction coefficients in order to exclude profiles, or parts of the profiles with potentially high uncertainties from their analysis.*

We conducted the intensive sensitivity tests using the simulated data of lidar and sky radiometer with random noises in the work of Kudo et al. 2016. The retrieval errors increased with a decrease of AOD or extinction coefficient. In the case that the AOD was about 0.05 and the extinction coefficient was about 0.01 km^-1, the retrieval errors were ±0.003 km^-1 for the extinction coefficient, and ±0.05 for single-scattering albedo and asymmetry factor. We added the explanation of the retrieval errors of the SKYLIDAR in Sect. 2.1.1 of the revised paper. And, we discussed our results of the optical properties and the retrieval errors in Sect. 3.1 of the revised paper.

Page 5, lines 8-25 in the revised paper.

*Specific Comments*

*Section 2.1.2 (page 4, lines 11-12)*

*Why do you use the wavelet covariance transform (WCT) at 532nm instead of 1064nm for the cloud and aerosol base retrievals when both are available during the day? The aerosol layers tend to appear more clearly in the 1064nm channel.*

In the beginning of this study, We used the 1064 nm channel, and many tests were conducted. However, the base height was sometimes too low. The signal noises in the 1064 nm channel were very large in the layers where the aerosol load was small. The influences of the signal noises remained in the vertical distribution of the WCT at 1064 nm and affected the estimation of the ABL height. On the other hand, the signal noises in the 532 nm channel were small. After the many tests, we decided to use the 532 nm channel at first. When the base height is not detected by the 532 nm channel, we try to detect the base height by the 1064 nm channel.

*Section 2.2.1 (page 5, lines 18-22)*

*It is mentioned here that "we determined the optical properties between 532 and 1064 nm by linear interpolation and used the optical properties at 532 and 1064 nm for wavelengths of less than 532 nm and greater than 1064 nm". Is this true for all the aforementioned optical properties? The aerosol extinction coefficient, in particular, usually has a strong wavelength dependence, even within the visible spectrum. Since the spectral range is much larger here, the authors should consider applying a correction, e.g. with the use of a constant angstrom exponent, for the extinction profiles in wavelengths below 532nm and above 1064nm.*

We calculated the real and imaginary parts of the refractive index between 532 and 1064 nm by the linear interpolation in a log-log plane. For wavelengths less than 532 nm and greater than 1064nm, the refractive index at 532 and 1064 nm were used. The extinction coefficient, single-scattering albedo, and phase function were calculated from these refractive indices, the volume size distributions, and the volume ratio of the non-spherical particles in the coarse mode. Then, the radiative fluxes and heating rates were calculated by the radiative transfer model. We revised the text.
Page 7, lines 16-25 in the revised paper.

*Section 3.1.1 (page 6, lines 22-23)*

*The authors mention that "In all the seasons, the extinction coefficient was large in three layers: from the surface to 1.5 km; from 1.5 to 3.5 km; and from 3.5 to 6 km altitude." The upper layer, however, only appears during winter and spring (Figure 2). Please rephrase. Since only the boundary layer is*

*provided in the figures the discussion should be focused mainly on the differences between the free atmosphere and the boundary layer.*

I revised the text.
Page 8, lines 28-31 in the revised paper.

*Section 3.1.1 (page 6, lines 31-32)*
*The number of daily mean profiles is considerably lower during summer (only 7 days - Figure 2). Is this related to cloudy or rainy conditions? The constant rainy conditions could explain the lower extinction values.*

Yes. Summer in Japan is hot and humid. The cumulus clouds develop almost every day. The small number of daily mean profiles is due to the cloudy or rainy conditions. We added this explanation in the text.
Page 8, lines 26-28 in the revised paper.

*Section 3.1.2 (page 8)*
*Please provide specific values from the analysis (Table 2). A short comparison with estimates from other scientific studies would be interesting here as well.*

We provided the specific values and compared them with other studies.
Pages 10-12 in the revised paper.

*Figure 4 The winter profiles exhibit high extinction and high SSA values while the summer profiles show low extinction and low SSA values. During spring both kinds of profile are observed. Do you observe any specific seasonal pattern within the spring months. For example, do the winter-like profiles tend occur in early spring while the summer-like profiles in late spring?*

We did not make the monthly profiles because the data number was not enough to make them. However, from the daily figures, the spring-like profile was observed sometimes in late winter. The observation period of two years was too short to investigate the monthly pattern.

*Technical Comments - Suggestions*

*Table 1*

*Consider defining abbreviations for the table properties in the caption and include the variability with +- in the same line with the mean value. Currently, each table property occupies 2 lines and sometimes this is difficult to follow.*

I revised the table.

Page 20 in the revised paper.

*Figure 6*

*A small height versus time plot per trajectory could be included along with the map. Regions where the trajectories pass closer to the ground are more probable to affect the layer aerosol content.*

I added the height versus time plot as Fig. 7d.

Page 30 in the revised paper.

*Section 3.1.1 (page 7, lines 14-15)*

*Add citation*

I added a citation.

Page 9, line 23 in the revised paper.

*Section 3.1.1 (page 7, lines 23-24)*

*typo - replace "almost same values" with "almost the same"*

We changed the contents in the paragraph to the comparisons of our results with other papers.

Page 10, lines 1-16.

[revised manuscript text omitted]

(a) Net radiation (W m⁻²) (b) Sensible heat flux (W m⁻²) (c) Latent heat flux (W m⁻²)

(d) Potential temperature (K) (e) Specific humidity (g kg⁻¹) (f) ABL Height (m)

Spring mean 2 April 2012 16 April 2013 8 May 2013 9 May 2013 14 May 2013

**Figure 8.** Results of EXP0 (dashed line) and the difference between EXP1 and EXP0 (solid lines): (a) net downward surface radiation, (b) sensible heat flux, (c) latent heat flux, (d) potential temperature at 12:00 LST, (e) specific humidity at 12:00 LST, and (f) ABL height.

[Figure]

**Spring mean**   2 April 2012   16 April 2013   8 May 2013   9 May 2013   14 May 2013

**Figure 9.** Results of EXP0 (dashed line) and the difference between EXP2 and EXP0 (solid lines): (a) net radiation at the surface, (b) sensible heat flux, (c) latent heat flux, (d) potential temperature at 12:00 LST, (e) specific humidity at 12:00 LST, and (f) ABL height.

[Figure]

**Figure 10.** Dependencies of the daily maximum 2 m temperature (a) and the daily maximum ABL height (b) on the aerosol optical thickness and Ångström exponent in the FA. The "ANG" indicates Ångström exponent. The color of filled circle indicates the value of Ångström exponent, from 0.0 to 0.5 (red), from 0.5 to 1.0 (orange), from 1.0 to 1.5 (green), and from 1.5 to 2.0 (blue). The solid lines are the results of the model simulations for the simplified aerosol vertical profile described in the text.